# Multi-omics analysis identifies an M-MDSC-like immunosuppressive phenotype in lineage-switched AML with *KMT2A* rearrangement

Takashi Mikami [1], Itaru Kato [1,21] ✉, Akira Nishimura [2], Minenori Eguchi-Ishimae[3], Tatsuya Kamitori [1], Keiji Tasaka[4], Hirohito Kubota [1], Tomoya Isobe [5], Yoshinori Uchihara[1], Yui Namikawa[2], Satoru Hamada[6], Shinichi Tsujimoto [7], Shotaro Inoue[8,9], Takayuki Hamabata[4], Kazushi Izawa [1], Takako Miyamura[10], Daisuke Tomizawa [11], Toshihiko Imamura[12], Hidemi Toyoda[13], Mariko Eguchi[3], Hiroaki Goto[14], Seishi Ogawa[15,16,17], Masatoshi Takagi [2], James Badger Wing [18,19,20,22] & Junko Takita [1,21,22] ✉

Lineage switching (LS) is the conversion of cancer cell lineage during the course of a disease. LS in leukemia cell lineage facilitates cancer cells escaping targeting strategy like CD19 targeted immunotherapy. However, the genetic and biological mechanisms underlying immune evasion by LS leukemia cells are not well understood. Here, we conduct a multi-omics analysis of patient samples and find that lineage-switched acute myeloid leukemia (LS AML) cells with *KMT2A* rearrangement (*KMT2A-r*) possess monocytic myeloid derived suppressor cell (M-MDSC)-like characteristics. Single-cell mass cytometry analysis reveals an increase in the M-MDSC like LS AML as compared to those of lineage-consistent *KMT2A-r* AML, and single-cell transcriptomics identify distinct expression patterns of immunoregulatory genes within this population. Furthermore, in vitro assays confirm the immunosuppressive capacity of LS AML cells against T cells, which is analogous to that of MDSCs. These data provide insight into the immunological aspects of the complex pathogenesis of LS AML, as well as development of future treatments.

Lineage switching (LS) refers to conversion of leukemia lineage during the course of disease, leading to relapse of disease and a poor clinical outcome[1]. LS used to be a very rare phenomenon, accounting for less than 1% of leukemia cases[1], but now presents as a serious complication in the context of emerging CD19 targeted immunotherapies such as bispecific T-cell engager, blinatumomab, and chimeric antigen receptor T cells (CAR-T cells); it is particularly problematic in the context of *KMT2A(MLL)* rearranged acute lymphoblastic leukemia (*KMT2A-r* ALL)[2–5]. In particular, since *KMT2A::AFF1* is the most common fusion among *KMT2A-r* ALL[6,7], the majority of lineage-switched acute myeloid

leukemia (LS AML) cases develop from *KMT2A::AFF1* rearranged ALL[2]. By contrast, de novo AML with *KMT2A::AFF1* fusion is rare, due primarily to the very strong propensity of *KMT2A::AFF1* to induce B cell precursor ALL rather than AML[6,8]. Several important papers on LS have been published in recent years, focusing mainly on the cells of origin and the mechanisms underlying lineage plasticity[9,10]; however, the characteristics of LS which could induce escape from immunotherapy are not fully understood.

Myeloid precursors differentiate into highly diverse cell types. Among these, some maintain an immature state, negatively

regulating immune responses to cancer and other diseases. This population, known as myeloid derived suppressor cells (MDSCs), inhibits proliferation of immune cells and induces production of regulatory T cells (Tregs)[11]. Over the past few years, the relevance of MDSCs to the immune dysregulation underlying hematological malignancies and allogeneic hematopoietic stem cell transplantation has been documented, and the potential role of MDSCs as biomarkers and therapeutic targets is attracting particular interest[12]. Additionally, in the field of AML, it is reported that a higher proportion of leukemic blasts showing an MDSC-like phenotype is associated with shorter overall survival[13].

In this study, we use multi-omics techniques, including RNA sequencing (RNA-seq), whole exome sequencing (WES), cytometry by time-of-flight (CyTOF), and single-cell RNA-seq to conduct a comprehensive analysis and gain deeper insight into disease pathogenesis. These multi-omics analyses reveal that LS AML with the *KMT2A* rearrangement contains an abundant fraction of AML cells with a monocytic MDSC (M-MDSC)-like phenotype. In addition, functional assays elucidate that LS AML cells suppress T cell proliferation and increase the percentage of effector Tregs, indicating that they have immunosuppressive properties analogous to those of MDSCs. These MDSC-like characteristics of LS AML may offer opportunities for the development of novel therapies targeting the immunological aspect of this highly refractory disease.

## Results

### Collection of lineage-switched AML and control samples, and data gathering

We collected five cases of LS leukemia that originally presented as B-cell precursor ALL and then relapsed as monocytic AML (hereafter referred to as LS AML) (Fig. 1a). The lymphoid or myeloid phenotypes were diagnosed clinically by morphology and lineage-specific marker expression. All LS AML cases were classified as M5 according to the French-American-British (FAB) classification. Four cases possessed the *KMT2A::AFF1* t(4;11)(q21;q23) rearrangement, and one possessed an aberrant truncated transcript of *KMT2A* containing intron 7 (Supplementary Fig. 1a–c). To avoid misclassifying secondary AML as LS AML, we confirmed that *KMT2A::AFF1* fusion breakpoints at the exon level were identical before and after LS (Supplementary Table 1). In addition, we collected five cases of monocytic *KMT2A-r* AML without LS, referred to hereafter as lineage-consistent (LC) AML: four cases had disease at presentation and one had relapsed. The characteristics of the in-house cohort, including information about sex, are summarized in Supplementary Table 2. Patients LS AML1 and LS AML3 received blinatumomab therapy before LS, and patient LS AML3 also received a CD19 CAR-T cell infusion (Tisagenlecleucel). Patient LS AML4 received hematopoietic stem cell transplantation before LS.

Sample collection for RNA-seq is summarized in Fig. 1b. To increase the sample size, we also leveraged deposited RNA-seq data. Four monocytic *KMT2A-r* LS AML data samples were obtained from the study reported by Tirtakusuma et al.[10], and 49 monocytic *KMT2A-r* LC AML data samples (40 at disease presentation, nine at relapse) were collected from the data matrix of the Therapeutically Applicable Research to Generate Effective Treatments (TARGET) AML initiative (https://www.cancer.gov/ccg/research/genome-sequencing/target)[14]. Information about the deposited samples is provided in Supplementary Table 3. Thus, the following transcriptomic analysis using the same in-house pipeline compared nine LS AML samples with 54 LC AML samples (44 at disease presentation and 10 at relapse) of the *KMT2A-r* monocytic lineage.

### Transcriptomic overview of lineage-switched AML within *KMT2A* rearranged AML

Principal component analysis (PCA) and hierarchical cluster analysis were performed to gain a transcriptomic perspective on LS AML. To

begin with, we conducted PCA to achieve dimensional reduction in order to better grasp the similarities in gene expression between each of the samples (Fig. 1c, d, Supplementary Fig. 2a, b). Possibly because all of the cases in this cohort had *KMT2A* rearrangements and shared a tendency to differentiate into monocytes (M4 or M5), the PCA plot revealed an ambiguous population rather than distinct clusters. LS AML samples tended to be located near to each other, and samples with the same fusion partner also tended to be located nearby. Since the PCA plot did not clearly distinguish LS AML, we conducted RNA sequencing-based gene expression clustering and found three consensus clusters within the *KMT2A-r* AML population (Fig. 1e). The cluster number was validated by stability evidence obtained from unsupervised analysis (Supplementary Fig. 2c–f). Cluster 1 was characterized by *KMT2A* fusion partners *MLLT3* and *MLLT10*, whereas Cluster 3 comprised mainly *ELL* and *MLLT4*. Interestingly, all LS AML samples were assigned to Cluster 2, which contained various other fusion partners. This may reflect a selection bias, as the majority of LS AML cases harbor *KMT2A::AFF1* fusion, which is also common in Cluster 2. To clarify the characteristics of LS AML, we used gene ontology analysis to make a detailed comparison between LS AML and LC AML in Cluster 2.

### Lineage-switched AML is characterized by downregulation of MHC class II molecules and possession of a monocytic-MDSC-like gene expression signature

The samples assigned to Cluster 2 by consensus clustering, including LS AML samples, were thought to have more common transcriptomic characteristics under the condition of various fusion partners. To identify differences in expression patterns between physiological pathways in LS AML and LC AML, we conducted Gene Set Enrichment Analysis (GSEA) to compare the LS AML samples with LC AML samples in Cluster 2[15,16]. Figure 2a shows the top 15 and bottom 15 enriched ontology gene sets, based on the normalized enrichment score (NES) (original data: Supplementary Data 1 and 2). Interestingly, gene sets related to adaptive immunity (purple) and antigen presentation (blue) accumulated in the bottom 15. This indicates that compared with LC AML in Cluster 2, LS AML was characterized by suppression of the acquired immune system and reduced antigen presentation, particularly in the context of major histocompatibility complex (MHC) class II. We also compared LS AML samples with whole LC AML samples by GSEA (Supplementary Fig. 3a, Supplementary Data 3 and 4). The gene sets related to adaptive immunity and antigen presentation were significantly downregulated in LS AML, similar to the comparison with Cluster 2. Notably, gene sets for antigen presentation were ranked even more prominently in this broader comparison.

We then analyzed the downregulated gene sets related to adaptive immunity and antigen presentation observed in LS AML. To detect common genes contributing to the low enrichment signals in each gene set group, we conducted leading edge analysis using the above two gene set groups ranked in the bottom 15 after GSEA of Cluster 2 (Fig. 2b, c). Within the gene sets related to adaptive immunity, *IRF4*, *GATA3*, and *HLA-DRB1* were particularly downregulated. With respect to gene sets related to antigen presentation, MHC class II genes (including HLA-DR [*HLA-DRA*, *DRB1*, and *DRB5*]) were downregulated. During differentiation of the monocytic lineage, expression of HLA-DR is usually observed as early as the granulocyte-monocyte progenitor stage[17]. By contrast, M-MDSCs are defined as monocytes with low or no expression of HLA-DR (i.e., $CD11b^+CD14^+CD15^-HLA-DR^{low/-}$)[11,18]. From this perspective, the immunosuppressive gene expression pattern, as well as downregulation of HLA-DR, in LS AML samples suggests that LS AML may be similar to M-MDSCs. In addition, deficiency of interferon regulatory factor 4 (IRF4) is reported to favor generation of MDSCs and increases expansion of M-MDSCs[19,20], which is consistent with this hypothesis.

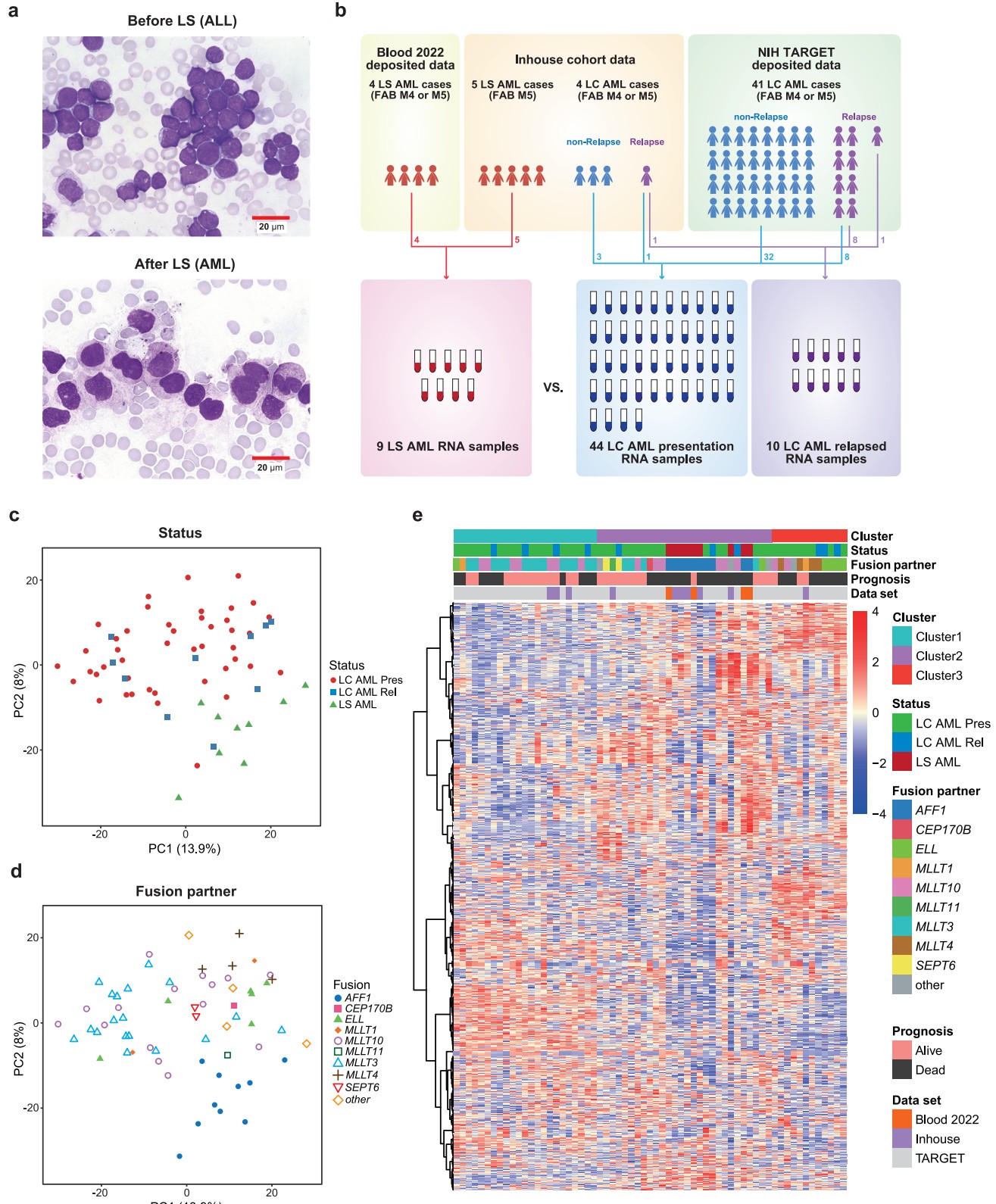

**Fig. 1 | Transcriptomic characterization of lineage-switched AML with the KMT2A rearrangement. a** Morphologic switching of LS from B-cell precursor ALL to monocytic AML (FAB: M5). Representative samples from patient LS AML2 are shown. May-Giemsa stain, ×400. **b** Schematic diagram showing sample collection and RNA-seq analysis of the inhouse cohort and deposited data. Most relapsed cases had samples taken at both disease presentation and relapse. **c, d** PCA distribution plots for each sample (Probe: 1500). The status (**c**) and fusion partner (**d**) of each sample are indicated. Pres: at disease presentation, Rel: at relapse. Source data are provided as a Source Data file. **e** Hierarchical clustering using Ward's distance (Pearson), based on the 1500 most differentially expressed genes across nine LS AML samples and 54 LC AML samples. The optimal cluster number was accessed by plots in ConsensusClusterPlus (Supplementary Fig. 2c–f).

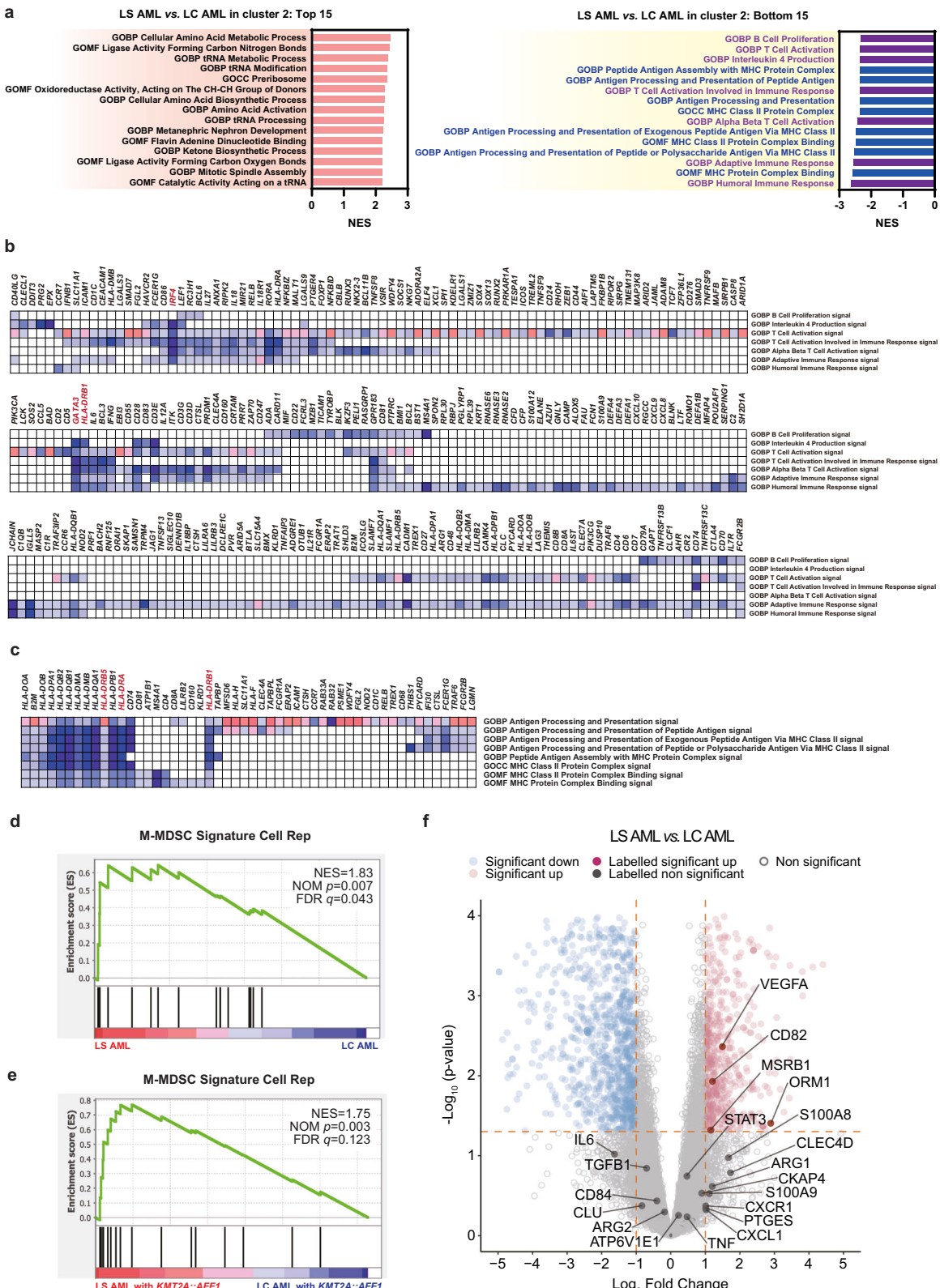

**a** LS AML *vs.* LC AML in cluster 2: Top 15 / LS AML *vs.* LC AML in cluster 2: Bottom 15

**b**

**c**

**d** M-MDSC Signature Cell Rep — NES=1.83, NOM *p*=0.007, FDR *q*=0.043

**e** M-MDSC Signature Cell Rep — NES=1.75, NOM *p*=0.003, FDR *q*=0.123

**f** LS AML *vs.* LC AML

To verify the hypothesis that LS AML is more similar to M-MDSC than LC AML, we compared the gene expression profiles with the M-MDSC gene signature reported previously. The "M-MDSC Signature Cell Rep" comprises the top 16 genes (*ORM1, S100P, SLCO4C1, CYP1B1, CLEC4D, CHD7, CKAP4, MSRB1, CLU, CD82, CPEB4, TLK1, OSBPL8, ABHD2, ATP6V1E1*, and *ROCK1*) that are significantly upregulated (*p* < 0.05) in M-MDSCs compared with classical monocytes[21]. When we compared the M-MDSC signature using GSEA, we found that it was significantly more enriched in LS AML than in LC AML (Fig. 2d). In addition, the M-MDSC signature was also enriched in LS AML with *KMT2A::AFF1* when compared with LC AML with *KMT2A::AFF1*, suggesting the possibility that its characteristics are derived from LS rather than from the *AFF1* fusion (Fig. 2e).

**Fig. 2 | Gene set enrichment analysis and analysis of differentially expressed genes in lineage-switched AML. a** Ontology gene sets ranked in the top 15 and bottom 15 normalized enrichment scores (NESs) calculated by gene set enrichment analysis (GSEA). LS AML and LC AML in Cluster 2 were compared. Adaptive immune-related gene sets are shown in purple; antigen presentation-related gene sets are shown in blue. GOCC gene ontology cellular component, GOBP gene ontology biological process, GOMF gene ontology molecular function, HP human phenotype. Source data are provided as a Source Data file. **b, c** Leading edge analysis of adaptive immune-related gene sets (**b**) and antigen presentation-related gene sets (**c**) ranked in the bottom 15 when comparing LS AML and LC AML in Cluster 2. Expression values are represented by colors, where the range of colors (red, pink, light blue, dark blue) reflects the range of expression values (high, moderate, low, lowest, respectively). **d** Enrichment analysis of M-MDSC signatures comparing LS AML and LC AML. The "M-MDSC Signature Cell Rep" comprises *ORM1, S100P, SLCO4C1, CYP1B1, CLEC4D, CHD7, CKAP4, MSRB1, CLU, CD82, CPEB4, TLK1, OSBPL8, ABHD2, ATP6V1E1*, and *ROCK1*. NOM: nominal; FDR: false discovery rate. Nominal (NOM) *p*-value represents the credibility of the enrichment result, and false discovery rate (FDR) *q*-value is the *p*-value adjusted after multiple hypothesis testing (Benjamini-Hochberg method, two-sided). **e** Enrichment analysis (pre-ranked) of M-MDSC signatures comparing LS AML with *KMT2A::AFF1* and LC AML with *KMT2A::AFF1*. NOM *p*-value represents the credibility of the enrichment result, and FDR *q*-value is the *p*-value adjusted after multiple hypothesis testing (Benjamini-Hochberg method, two-sided). **f** Volcano plot showing differentially expressed genes (DEGs) between LS AML and LC AML. The significance-adjusted *p*-value (calculated by DESeq2 using multiple comparisons; Benjamini-Hochberg method, two-sided) is set as <0.05, and a significant-fold change is defined as > |2| (orange dotted lines). The names of the M-MDSC-related genes are displayed as dark red or gray dots.

Furthermore, we examined differentially expressed genes (DEGs) between LS AML and LC AML (Fig. 2f, Supplementary Data 5). Figure 2f shows genes related to M-MDSCs[18,21,22] (Supplementary Table 4), along with their names. Among M-MDSC-related genes, expression of *VEGFA, CD82, ORM1*, and *MSRB1* was upregulated significantly in LS AML. These results indicate that, compared with that of LC AML, the gene expression pattern of LS AML is more similar to that of M-MDSCs.

As a side note, WES of our in-house LS AML cases identified several mutations (Supplementary Data 6). Among those, high variant allele frequencies (VAFs) of *TP53* in LS AML1 and LS AML3, and of *CDKN2A* in LS AML2 were attribute to copy number alteration (Supplementary Fig. 3b).

## Mass cytometry analysis identified an M-MDSC phenotype enriched in lineage-switched AML cells

The RNA-seq results indicate that the characteristics of LS AML samples are more similar to M-MDSCs than are those of LC AML samples; however, bulk tumor RNA-seq data can be affected by non-tumor cells within the analyzed sample. To verify whether the observed characteristics are due to AML cells themselves or to normal (non-AML derived) M-MDSCs, and to confirm expression of MDSC-specific surface antigens at the single-cell level, we designed a mass cytometry panel comprising 39 markers (Supplementary Table 5), which can be used to analyze leukemic cells and immune cells simultaneously.

To begin with, the obtained CyTOF data were dimensionally reduced using 39 multi-parameters and then visualized by viSNE, a visualization tool based on the t-distributed Stochastic Neighbor Embedding (t-SNE) algorithm[23]. The viSNE plots clearly separated AML cells (pink) from other normal immune cells (Fig. 3a). Next, using the same data, we performed conventional gating to identify cells showing an M-MDSC phenotype (i.e., CD11b⁺CD14⁺CD15⁻HLA-DR^low/-), without distinguishing AML cells or other normal cells (Supplementary Fig. 4a). When setting the gates, peripheral blood mononuclear cells (PBMCs) from a healthy individual, which contained 2.3% CD11b⁺CD14⁺CD15⁻HLA-DR^low/- cells, were used as a reference. Previous reports show that the percentage of M-MDSCs within the PBMC population of a healthy individual is about 0.5–3%[24–26]. Finally, identified cells with a M-MDSC phenotype were overlaid onto the viSNE plot (shown in red) to determine if they were AML cells (Fig. 3a)[27,28]. The identified cells with a M-MDSC phenotype were largely AML cells. A representative viSNE plot of LS AML1 cells is shown with expressed M-MDSC-related markers (Fig. 3b). Notably, the percentage of M-MDSC-like AML cells within the total AML cell population was significantly higher in LS AML samples (Fig. 3c). Furthermore, the M-MDSC-like AML population in patient LS AML2 increased after 70 days post-LS onset (i.e., at the time of exacerbation), suggesting that this population may be involved in disease progression (Fig. 3a). These results demonstrate that some AML cells possess M-MDSC-like properties, and that they are more common in LS AML than in LC AML. In addition, we evaluated expression of HLA-DR by leukemia cells from each patient before and after LS (Supplementary Fig. 4b). In all patients, expression of HLA-DR by ALL cells was >95%, and decreased after LS; this indicates that downregulation of HLA-DR expression occurs after development of LS.

In addition, the proportion of Tregs (CD4⁺CD25⁺CD127⁻ T cells) and their effector subsets (CD4⁺CD25⁺CD127⁻CD45RA⁻Foxp3^high T cells[29,30]) within the CD4⁺ T cell population was higher in LS AML samples than in LC AML samples (Fig. 3d, e). The gating strategy used to identify Tregs and their effector subsets is shown in Supplementary Fig. 5.

Taken together, the increase in the M-MDSC-like AML population, as well as augmentation of Treg numbers within the CD4⁺ T cell population, suggest an immunosuppressive environment in patients with LS AML.

Next, we detected *KMT2A::AFF1* in samples LS AML1, 2, 3, and 4 (which contained an adequate number of viable cells for downstream analysis) to verify that the M-MDSC-like population comprised leukemic rather than non-AML-derived MDSC cells. Initially, AML cells in each patient sample were gated as a CD45^dim population using conventional CD45/SSC gating, followed by sorting of the CD11b⁺CD14⁺CD15⁻HLA-DR^low/- fraction within the AML cells (Supplementary Fig. 6). RNA-seq detected the *KMT2A::AFF1* fusion within the M-MDSC-like population of LS AML1 and LS AML2 (Supplementary Table 1). Since the number of M-MDSC-like AML cells in LS AML3 and LS AML4 was small, we identified the break point of the *KMT2A::AFF1* fusion sequence directly by nested polymerase chain reaction (PCR) using DNA extracted from M-MDSC-like AML cells. The primers used for the PCR analysis were designed based on primary ALL samples from each patient. The sequence of the PCR product obtained from M-MDSC-like AML cells was identical to that obtained from primary ALL cells (Supplementary Fig. 7).

## Single-cell transcriptomics identified M-MDSC-like AML cells, and characteristic gene expression patterns related to regulation of the immune system

Mass cytometry analysis identified some LS AML cells with the M-MDSC phenotype. This indicates that each leukemic cell from a patient may show different levels of similarity to M-MDSCs due to heterogeneity. Therefore, we used a single-cell transcriptomic approach to better understand the genetic characteristics of M-MDSC-like AML cells.

First, CD45^dim viable leukemic blasts were sorted from the LS AML and LC AML samples by flow cytometry, followed by sequencing and curation. The single-cell data obtained from AML cells were then dimensionally reduced by Uniform Manifold Approximation and Projection (UMAP; arXiv: 1802.03426), along with data from BM mononuclear cells obtained from three pediatric patients with autoimmune diseases (systemic lupus erythematosus: SLE, systemic juvenile idiopathic arthritis: sJIA, Sjogren syndrome; Fig. 4a). The patients with autoimmune diseases had chronic inflammation, which promotes

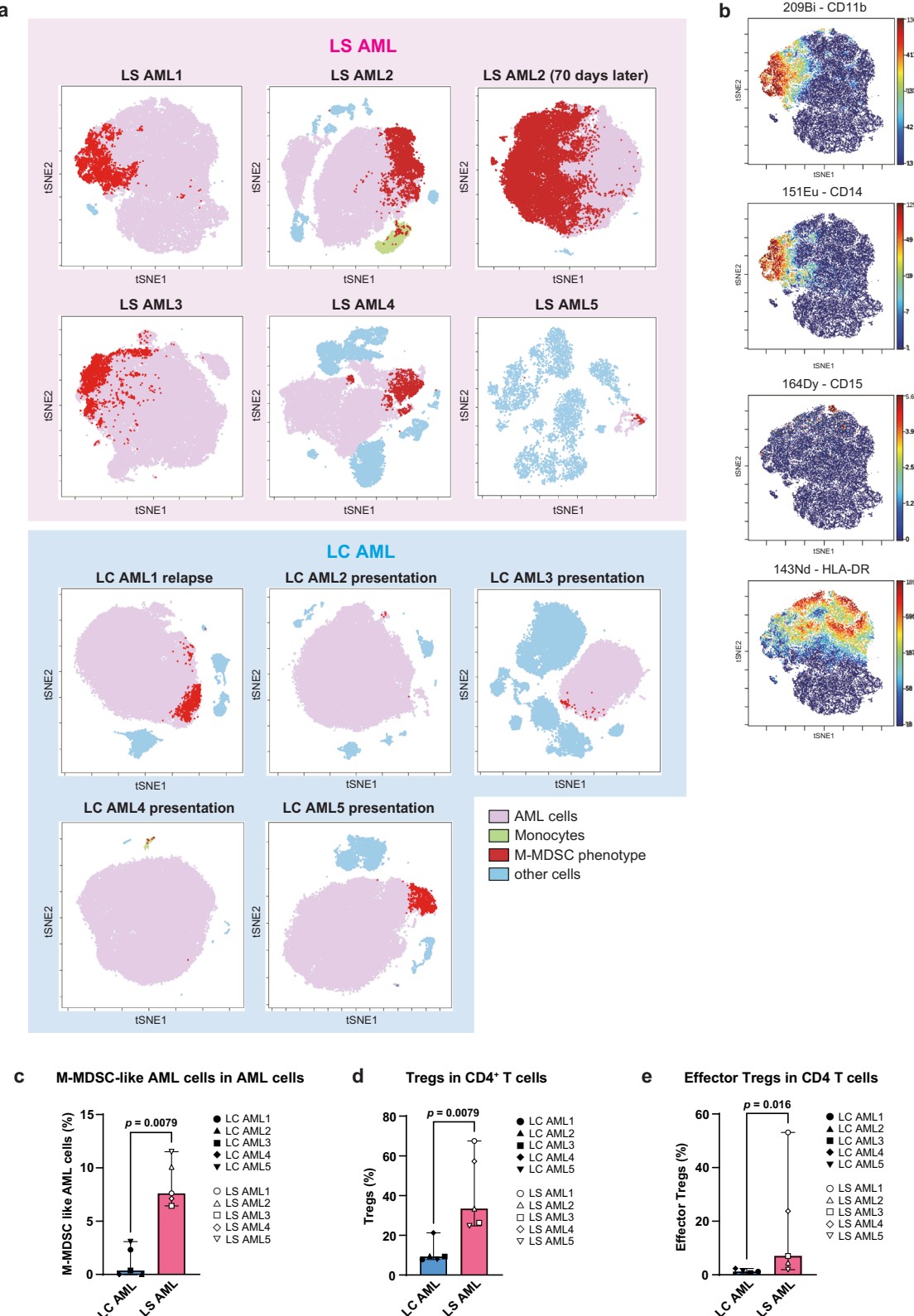

generation of MDSCs[31]. As expected, AML cells with a monocyte signature and lacking expression of *HLA-DR* were enriched with the M-MDSC signature[18] (Fig. 4b).

Subsequently, we classified AML cells as either "M-MDSC-like" or "non-M-MDSC-like", according to the normalized expression level of M-MDSC signature genes (Fig. 4c), and then used Metascape software to compare the M-MDSC-like LS AML subset with the non-M-

MDSC-like LS AML subset with respect to enrichment of gene sets associated with hematopoiesis[32]. In the M-MDSC-like LS AML subset, many gene sets related to immunity, including pathways that regulate immune responses, cytokine production, and immune effector processes, ranked highly (Fig. 4d). Furthermore, the upregulated and downregulated gene sets observed during the comparison were similar to those observed during comparison of M-MDSCs and non-

**Fig. 3 | Detection of M-MDSC-like AML cells and Tregs in AML samples by mass cytometry. a, b** Mass cytometry viSNE plots showing expression of 39 markers. Cells consistent with the definition of M-MDSCs (i.e., CD11b⁺CD14⁺CD15⁻HLA-DR$^{low/-}$) are red, AML cells are pink, monocytes are yellow-green, and other immune cells are light blue (**a**). The LS AML1 sample is shown as a representative plot, and expression of CD11b, CD14, CD15, and HLA-DR is displayed as a heatmap (**b**). **c** Percentage of M-MDSC-like AML cells within the AML cell population. LC AML cases ($n = 5$) and LS AML cases ($n = 5$) were compared (median with 95% confidence interval (CI); statistical analysis was performed using the Mann-Whitney test, two-sided, $p = 0.0079$). **d** Percentage of Tregs (CD4⁺CD25⁺CD127⁻ T cells) within the CD4⁺ T cell population. LC AML cases ($n = 5$) and LS AML cases ($n = 5$) were compared (median with 95% CI; $p = 0.0079$, Mann-Whitney test, two-sided). **e** Percentage of effector Tregs (CD4⁺CD25⁺CD127⁻CD45RA⁻Foxp3$^{high}$ T cells) within the CD4⁺ T cell population. LC AML cases ($n = 5$) and LS AML cases ($n = 5$) were compared (median with 95% CI; $p = 0.016$, Mann-Whitney test, two-sided). Source data of Fig. 3c-e are provided as a Source Data file.

M-MDSC myeloid cells in patients with autoimmune diseases (Supplementary Fig. 8), indicating the immunological similarity of M-MDSC-like LS AML cells and M-MDSCs. To further capture the relationships among gene set characteristics of the M-MDSC-like LS AML subset, we selected a subset of enriched terms and rendered them as a network plot using Cytoscape[33]; in the network plot, terms associated with more genes had a more significant $p$-value. In addition, terms involved in regulation of the immune system and cell activation were located centrally and formed a dense network within the plot (Fig. 4e). The network of enriched terms emphasized the commitment of the M-MDSC-like AML subset to the immune system, as well as involvement in its regulation.

To estimate the homology between LS AML cells and M-MDSCs in individuals with autoimmune diseases, we performed a differentiation trajectory analysis using the SPRING pipeline[34,35]. The resulting graph revealed putative differentiation trajectories, including a trajectory depicting a continuum in which cells developed from hematopoietic stem cells (HSCs)/multipotent progenitors (MPPs) to M-MDSCs (Fig. 4f). In the trajectory map, M-MDSCs from patients with autoimmune diseases were located on the upper-right side, and were characterized by the enriched signals with an M-MDSC signature. Interestingly, M-MDSC-like LS AML cells did not show much overlap with the area in which M-MDSCs resided, although they were located along a similar trajectory (lower-right side of the map). The possible reason for the dissociation of the two populations is that M-MDSC-like LS AML cells express a neutrophil-myeloid progenitor signature, which is not observed in M-MDSCs.

The results of single-cell RNA-seq revealed that not only did M-MDSC-like LS AML cells have an immunological gene expression profile similar to that of M-MDSCs, but they also had some differences with respect to hematopoietic differentiation.

### Lineage-switched AML cells suppress T cell proliferation and induce Tregs in in vitro assays, suggesting an immunosuppressive capacity analogous to that of MDSCs

Functional evaluations are necessary to confirm the presence of MDSCs. For the next step, we asked whether LS AML cells have the same immunosuppressive properties as MDSCs, including T cell suppression and enhancement of Tregs.

First, we performed a T cell suppression assay. Responder CD4⁺CD25⁻CD45RA⁺ T cells (naïve Tconv) were sorted (purity > 95%) from healthy donor PBMCs using a FACSAria II cytometer (BD), stained with carboxyfluorescein succinimidyl ester (CFSE), and then co-cultured with different numbers of AML cells in the presence of IL-2 and CD3/CD28 beads. As a positive control for T cell suppression, we induced M-MDSCs from healthy donor PBMCs by culturing them in medium supplemented with IL-6, GM-CSF and isoproterenol, as described by Choi et al.[36]. After co-culture, the number of dividing T cells was counted, and production of interferon-γ (IFN-γ) by T cells was measured on the fifth day by intracellular staining (Fig. 5a). As expected, LS AML cells inhibited T cell proliferation in a dose-dependent manner, and to a much greater extent than LC AML cells (Fig. 5b, c). Moreover, the amount of IFN-γ produced by T cells decreased as the proportion of LS AML cells increased, in contrast to observations for LC AML cells (Fig. 5b, d). These data suggest that LS AML cells suppress IFN-γ production by T cells in a dose-

dependent manner. We obtained identical results when co-culturing naïve Tconv with induced M-MDSCs (Fig. 5c, d).

Second, we asked whether LS AML cells increase the Tregs population. CD4⁺ T cells isolated from a healthy individual by magnetic sorting were co-cultured for 5 days with AML cells or induced M-MDSCs (ratio = 1:1). Then, the percentage of Tregs and effector Tregs within the CD4⁺ T cell population was calculated (Fig. 5e). The gating strategy used to define Tregs (CD4⁺CD25⁺CD127⁻ T cells) and effector Tregs (CD4⁺CD25⁺CD127⁻CD45RA⁻Foxp3$^{high}$ T cells; Fraction II) was based on CD4⁺ T cells isolated from healthy control PBMCs, as described previously (Fig. 5f)[29,30]. Notably, co-culture with LS AML cells led to a significant increase in the percentage of effector Tregs, as did co-culture with induced M-MDSCs, and both were superior to co-culture with LC AML cells (Fig. 5g). Taken together, these data suggest that LS AML cells suppress T cell proliferation and increase the percentage of effector Tregs, indicating that they have immunosuppressive properties analogous to those of MDSCs.

Finally, we performed an in vitro high-throughput drug sensitivity test (DST) to screen LS AML[37]. Thirty anti-leukemic drugs were used for the DST; these drugs included conventional AML and ALL therapeutic drugs, and several chemotherapeutics used to reduce MDSC numbers in previous reports[38–40] (Supplementary Table 6). After 4 days of culture in the presence of serially diluted drugs, we measured the viability of AML cells and calculated the drug effect score (DES)[37]. LS AML samples were sensitive to gemcitabine, docetaxel, and PIK3/mTOR inhibitors (Supplementary Fig. 9). By contrast, agents with greater specificity for MDSCs, such as anti-S100A8/A9 and anti-CD93 antibodies, did not exhibit notable efficacy. However, the small number of samples and the lack of in vivo validation mean that further studies are needed to identify drugs that target LS AML effectively.

## Discussion

Here, we used a single-cell, multi-omics approach to comprehensively characterize LS AML. Our analysis revealed that LS AML cells with *KMT2A-r* contain a subpopulation with gene expression profiles, protein characteristics, and functional behaviors similar to those of M-MDSCs.

In the context of immune escape, it is suggested that AML sometimes has an effect on other immune cells within the body[41,42]. For example, AML cells overexpress inhibitory T cell ligands, induce NK cell dysfunction, and downregulate HLA expression to "hide" from immune cells; they also release humoral factors that inhibit proliferation of T and NK cells[42–45]. In addition, AML takes advantage of the immunosuppressive environment by inducing MDSCs[12]. Several clinical trials have targeted MDSCs in AML, although they are still less common than trails for other hematological malignancies such as myelodysplastic syndromes[12,46]; however, the types or subgroups of AML that are more capable of escaping immune surveillance are unclear.

Recently, researchers have begun to evaluate AML cells in terms of their similarity to MDSCs. Hyun et al. analyzed CD11b⁺CD33⁺HLA-DR⁻ MDSC-like blasts from patients with AML, and found that MDSC-like blasts showed higher expression of arginase I and inducible nitric oxide synthase, significantly suppressing CD8⁺ T cell proliferation induced by phytohemagglutinin A. They also reported that patients with a high number of MDSC-like blasts at the time of diagnosis

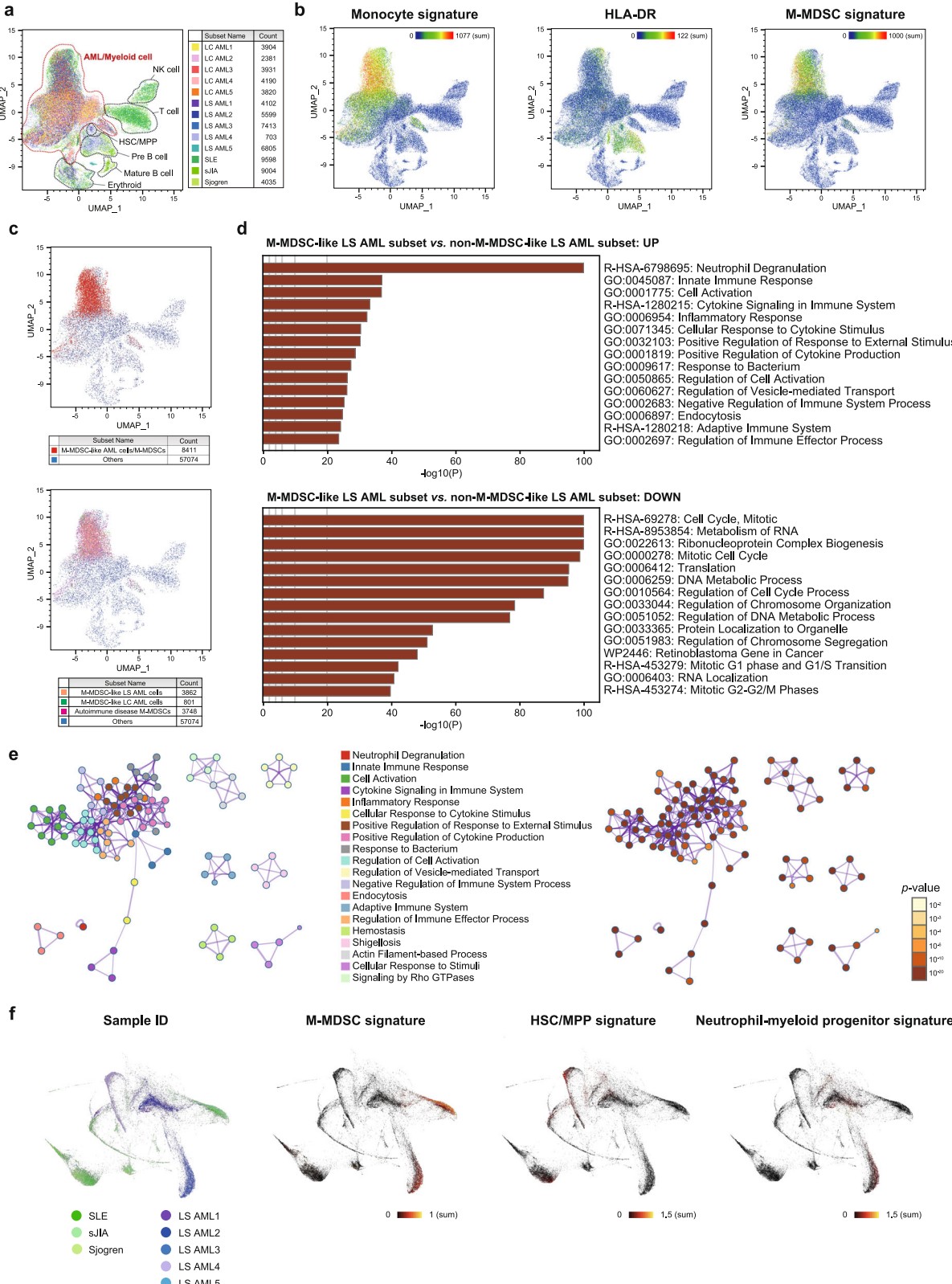

showed shorter overall survival[13]. Interestingly, Varela et al. succeeded in generating M-MDSCs from the monocytic myeloid leukemia cell line THP-1 in vitro by stimulating the latter with G-CSF and IL-4. Supernatant from the generated M-MDSCs inhibited proliferation of activated lymphocytes and impaired NK cell-induced apoptosis of leukemia cells[47]. As mentioned above, certain AML cells showed the same phenotype and behavior as MDSCs, and influenced the surrounding immune environment; these data align with our findings reported herein.

It remains unclear why *KMT2A-r* LS AML harbors a higher proportion of MDSC-like cells than LC AML. The *KMT2A* gene encodes a histone-H3 lysine-4 (H3K4) methyltransferase involved in epigenetic regulation of hematopoietic stem cell development[48]. Recently, Chen et al. identified a hematopoietic stem and progenitor cell-like (HSPC-

**Fig. 4 | Single-cell RNA-seq analysis of lineage-switched AML compared with M-MDSCs in autoimmune diseases. a** UMAP plot of single-cell transcriptomic data obtained from five LS AML, five LC AML, and three autoimmune disease samples. SLE systemic lupus erythematosus, sJIA systemic juvenile idiopathic arthritis. **b** The normalized intensity of the gene signatures (monocyte, HLA-DR, and M-MDSC) on the UMAP is represented by color. **c** The M-MDSC-like AML subset and M-MDSCs (red) are defined as cells with a normalized expression level of M-MDSC signature genes ≥ 150 (upper figure). Within that subset, LS AML cells are shown in orange, LC AML cells in green, and autoimmune disease M-MDSCs in red (lower figure). **d** The M-MDSC-like LS AML subset and the non-M-MDSC-like LS AML subset were compared by hematopoietic gene set analysis. Enriched terms across input gene lists are ranked according to their adjusted *p*-value. A two-sided well-adopted hypergeometric test and Benjamini-Hochberg *p*-value correction were used for multiple comparisons. R-HSA reactome-Homo sapiens, GO gene ontology, WP WikiPathways. Source data are provided as a Source Data file. **e** Network of terms enriched in gene sets characteristic of the M-MDSC-like LS AML subset. In the left figure, these are colored according to cluster ID; nodes that share the same cluster ID are typically close to each other. In the right figure they are colored by *p*-value. A two-sided well-adopted hypergeometric test and Benjamini-Hochberg *p*-value correction were used for multiple comparisons. **f** k-nearest neighbor graphs created by the SPRING pipeline. The expression of each gene signature (M-MDSC, HSC/MPP, and neutrophil-myeloid progenitor) is shown by color.

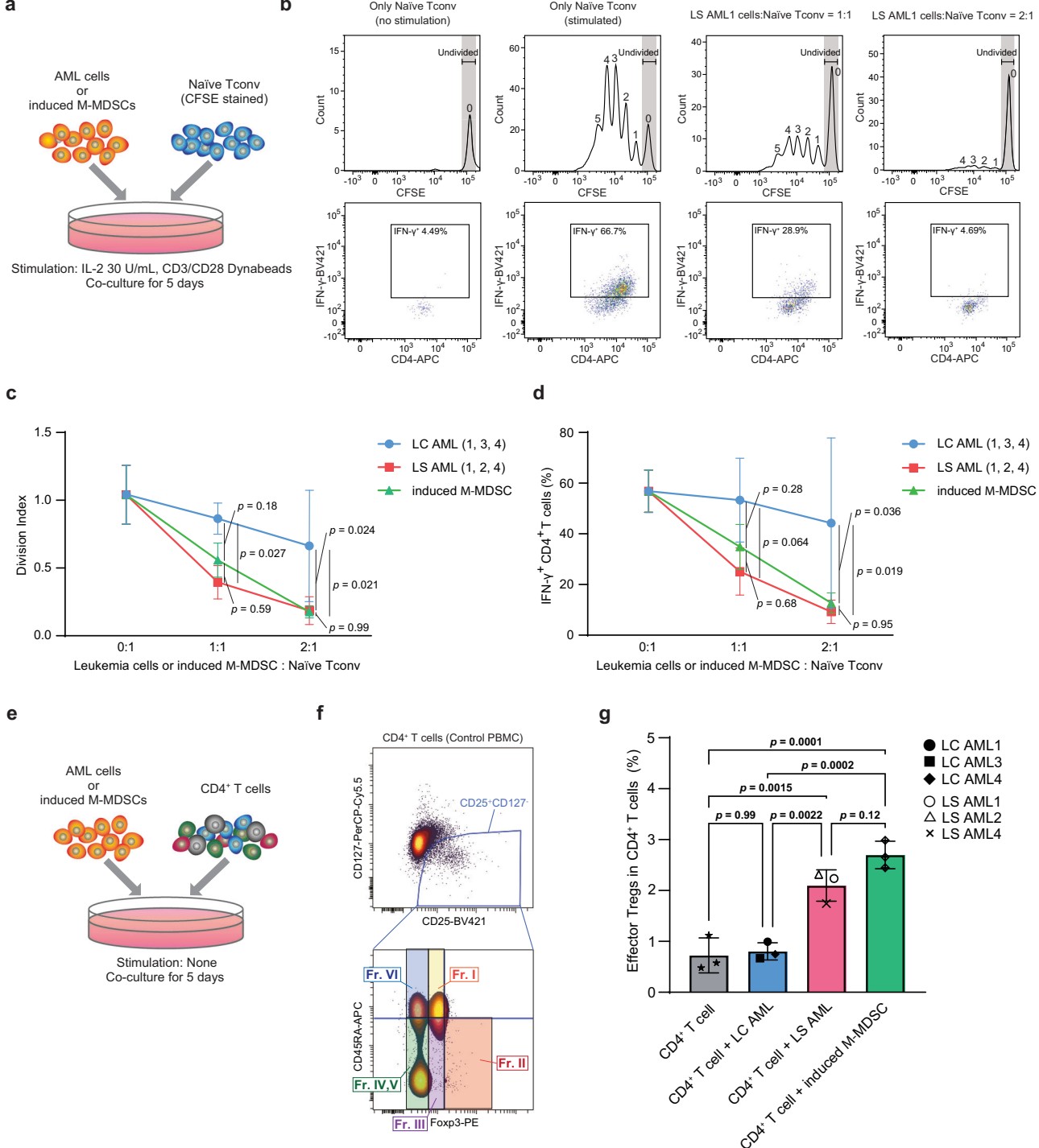

**Fig. 5 | Functional in vitro co-culture assays validating the MDSC-like characteristics of lineage-switched AML.** AML cells from patients LC AML1, 3, 4, and from patients LS AML1, 2, 4, were used for the in vitro co-culture assays. **a** Schematic diagram of the T cell suppression assay using AML cells and induced M-MDSCs. **b** Representative CFSE histograms and dot plots of IFN-γ intracellular staining obtained after 5 days of culture in the absence of LS AML1 cells, or in the presence of LS AML1 cells (at various ratios). The number of T cell divisions is indicated above each peak. **c** The T cell division index at different [AML cell or induced M-MDSC]/T cell ratios (mean ± standard deviation (SD); statistical analysis was performed using two-way ANOVA with Tukey's multiple comparisons test). LC AML cells (from patients LC AML1, 3, 4: $n = 3$), LS AML cells (from patients LS AML1, 2, 4: $n = 3$), and induced M-MDSCs made from PBMC from different healthy individuals ($n = 3$) were compared. **d** Percentage of IFN-γ$^+$ CD4$^+$ T cells within the total CD4$^+$ T cell population at different [AML cell or induced M-MDSC]/T cell ratios (mean ± SD; statistical analysis was performed using two-way ANOVA with Tukey's multiple comparisons test). LC AML cells (from patients LC AML1, 3, 4: $n = 3$), LS AML cells (from patients LS AML1, 2, 4: $n = 3$), and induced M-MDSCs made from PBMC from different healthy individuals (n = 3) were compared. **e** Schematic diagram of the Treg co-culture assay. **f** The gating criterion used to identify effector Tregs within CD4$^+$ T cells in control PBMCs. The CD25$^+$CD127$^-$CD45RA$^-$Foxp3$^{high}$ population (Fraction II), located in the bottom right part of the plot, represents effector Tregs; the same gating scheme was adopted for all co-culture samples. Fr: fraction. **g** The percentage of effector Tregs within the CD4$^+$ T cell population after 5 days of co-culture (mean ± SD; one-way ANOVA with Turkey's multiple comparison test). LC AML cells (from patients LC AML1, 3, 4: $n = 3$), LS AML cells (from patients LS AML1, 2, 4: $n = 3$), and induced M-MDSCs made from PBMC from different healthy individuals ($n = 3$) were compared. All experiments above were repeated and yielded similar results. Source data of Figs. 5c-d, and 5g are provided as a Source Data file.

like) population in the blood of patients with *KMT2A-r* ALL. This HSPC-like population formed an immunosuppressive signaling circuit with cytotoxic lymphocytes through the interferon γ pathway, thereby facilitating immune escape of leukemic cells[9]. Because LS AML sometimes arises directly from an HSPC-like population[10], it might be possible that LS AML has similar immunosuppressive capacities. Our GSEA results reveal that reduced expression of *IRF4* is a leading factor that downregulates gene sets related to acquired immune responses in LS AML to a greater extent than in LC AML. The transcription factor IRF4 is required for normal function and development of lymphocytes and macrophages, particularly in the context of T-cell receptor (TCR) signaling[49]. IRF4 exerts oncogenic effects in a lineage- and stage-specific manner: it acts as a tumor suppressor in immature lymphoid and myeloid neoplasms, but is overexpressed by various mature lymphoid neoplasms[50]. Wang et al. used a zebrafish embryo model to show that IRF4 prevents conversion of T lymphoid-primed progenitors to myeloid progenitors by repressing Pu.1[51]. In addition, expression of IRF4 in the tumor immune environment is suppressed during development of MDSCs, as well as during tumor formation[19]. These findings suggest that IRF4 has the potential to alter the developmental fate (e.g., lineage switching) of hematopoietic progenitors, and its downregulation may endow immature myeloid malignancies with MDSC-like properties.

Treatment-induced selection pressure can, on occasion, confer a survival advantage on certain leukemia subpopulations. LS is an increasingly common complication in the context of immunotherapies[5], the efficacy of which may be affected negatively by immunosuppressive cells such as MDSCs and Tregs. From this point of view, our observation of increased numbers of MDSC-like blasts and Tregs after LS may be interpreted as a new form of therapeutic resistance, which allows leukemia cells to evade T cell attack induced by immunotherapies. Therefore, targeting M-MDSC-like blasts, in addition to conventional leukemia treatment, could improve the efficacy of treatments for LS AML. However, even to this day, strategies to eliminate MDSCs or MDSC-like blasts in cases of hematological malignancy remain limited due to the complexity of identifying ideal markers, as well as a lack of specific approaches to targeting them in an immune environment that changes constantly when subjected to multidisciplinary leukemia therapies[52].

This study has some limitations. The main limitation is the small number of samples that could be analyzed. Particularly for the drug sensitivity test, the small number of analyzed cases did not allow statistical analysis. When we analyzed Tregs numbers in clinical samples, we found that the total number of CD4$^+$ T cells was small (3.7% on average) due to expansion of leukemia cells. Considering the rarity of effector Tregs within CD4$^+$ T cell populations, ~$5 \times 10^5$–$1 \times 10^6$ measured cell events are desirable to obtain a coefficient of variation less than 10%[53]; however, this was sometimes difficult to achieve in our CyTOF analysis. Therefore, although we revealed the characteristics of

a small number of LS AML cases by conducting a multi-omics analysis, a confirmatory analysis based on a larger number of cases is warranted. Another limitation is that most of the LS AML cases analyzed had the *KMT2A::AFF1* fusion. This is because *KMT2A-r* ALL develops into LS far more often than ALL with other translocations[2,3], and *KMT2A::AFF1* is the most common fusion among *KMT2A-r* ALL[6,7]; therefore, the MDSC-like features should also be assessed in LS AML cases with other genetic backgrounds. However, since LS AML with *KMT2A-r* is the most common disease entity in LS, the importance of paying close attention to MDSC-like features after development of LS remains the same. In addition, although our co-culture assays revealed that LS AML blasts have immunosuppressive capacity, it would be preferable to use sorted M-MDSC-like AML blasts for co-culture (when the number of viable AML cells in a sample is sufficient).

In conclusion, the multi-omics analysis conducted herein revealed that LS AML with the *KMT2A* rearrangement is characterized by an M-MDSC-like phenotype and immunosuppressive capacity. These MDSC-like characteristics can be exploited for future development of novel therapies, thereby helping to overcome this highly refractory disease.

## Methods
### Sample collection and preservation
Mononuclear cells obtained from the patient's peripheral blood or BM by density centrifugation were frozen in liquid nitrogen until required. Lymphoprep (Serumwerk Bernburg, cat: 18061) and CELLBANKER 1 (ZENOAQ RESOURCE, cat: 11910) were used for density centrifugation and viably preservation, respectively. The inhouse samples contained an average of 87.5% leukemia cells, and whole mononuclear cells were used for bulk RNA sequencing and whole exome sequencing.

### RNA extraction, sequencing, and subsequent analysis
RNA isolation was carried out using an RNeasy Mini Kit (QIAGEN, cat: 74104), according to the manufacturer's instructions. Agilent 4150 TapeStation and RNA Screen Tapes (Agilent Technologies, cat: 5067-5576) were used to measure RNA integrity, and libraries were prepared using the NEBNext Ultra II RNA Library Kit for Illumina (New England Biolabs, cat: E7770L) and MGIEasy Circularization Kit (MGI Tech, cat: 1000005259). Constructed single-stranded circular DNA libraries were run on DNBSEQ-G400RS (MGI Tech) in 150-basepair (bp) paired-end mode. Sequence alignment to the GRCh37 human genome assembly was performed using STAR[54], and read counting and downstream data analysis were performed using the in-house pipeline Genomon v.2.6.2 (https://github.com/Genomon-Project/). The deposited data were also processed using this pipeline. Fusion transcripts were detected and filtered by excluding fusions (i) mapping to repetitive regions; (ii) with fewer than four spanning reads; (iii) that occurred out of frame; or (iv) had junctions not located at known exon–intron boundaries. The expression level of each RefSeq gene was calculated from mapped

read counts and normalized using the R package DESeq2 version 1.36.0[55].

PCA distribution plots were created by ClustVis[56], using 1500 probes. Hierarchical consensus clustering was performed using the R package ConsensusClusterPlus[57], with 1000 iterations to ascertain cluster stability. The batch of samples was adjusted during the clustering process. Heat maps were generated by R package pheatmap. Alignment and clustering were performed by the supercomputer SHIROKANE at Human Genome Center, Institute of Medical Science, The University of Tokyo. DEGs were extracted using the R package DESeq2 and presented as volcano plots by R package ggvolcanoR[58].

### Gene set enrichment analysis

Gene set enrichment was analyzed by GSEA software (v 4.3.2) from the Broad Institute[15,16]. Differentially regulated MSigDB ontology gene sets (c5.all.v2022.1) with a false discovery rate $q$ value < 0.1 were filtered and evaluated. The gene set "M-MDSC signature" was manually constructed in accordance with previous reports[21], and analyzed with the ontology gene sets. The "M-MDSC Signature Cell Rep" comprised the 16 genes (*ORM1, S100P, SLCO4C1, CYP1B1, CLEC4D, CHD7, CKAP4, MSRB1, CLU, CD82, CPEB4, TLK1, OSBPL8, ABHD2, ATP6V1E1*, and *ROCK1*)[21].

### Single cell RNA extraction and sequencing

The BD Rhapsody Single-Cell Analysis System (BD Rhapsody Whole Transcriptome Analysis (WTA) Amplification Kit, BD Biosciences, cat: 633801) was used for scRNA-seq. CD45[dim] viable leukemic blasts were sorted by flow cytometry and tagged for multiplex analysis using the BD Human Single-Cell Multiplexing Kit (BD Biosciences, cat: 633781). Cell lysis, complementary DNA synthesis, and library construction were performed according to the manufacturer's instructions. Subsequently, sequencing was conducted using the HiSeq X platform (Illumina), with 150-bp paired-end reads. Paired-end scRNA-seq reads were processed according to the BD Rhapsody WTA bioinformatics workflow (BD Biosciences) on the Seven Bridges Genomics cloud platform, and aligned using STAR[54]. The cell barcode and unique molecular index (UMI) were identified, and recursive substitution error correction (RSEC) was calculated counts (as the final molecular counts) by removing the effect of UMI errors. RSEC counts were utilized for downstream analysis. The gene counts for each cell were normalized as counts per 10,000 reads. Cells were excluded based on a quality check (QC): ≥ 20% mitochondrial reads, and less than 10 expressed genes or ≥ 2000 expressed genes. For each dataset, PCA was conducted using the most variable genes. UMAP using the PCA data was conducted to reduce dimensionality. Unsupervised pathway analysis was carried out using Metascape software (https://metascape.org/gp/index.html#/main/step1)[32]. The gene sets comprising hematopoietic signatures were the same as those used in a previous report[59]. SPRING was used to perform a trajectory analysis[34]. Cells were visualized in K-nearest-neighbor (KNN) graphs based on the SPRING pipeline. The normal hematopoietic differentiation trajectory was visualized using BM mononuclear cells isolated from patients with autoimmune disease, and AML cells were overlaid onto this trajectory map using the subplot method. The processing parameters used by SPRING are shown in Supplementary Table 7.

### DNA extraction, whole exome sequencing, and mutation calling

DNA was isolated from preserved mononuclear cells using a QIAamp DNA Mini Kit (QIAGEN, cat: 51304), and the amount of double-stranded DNA was measured in a Qubit 4 Fluorometer (Invitrogen, cat: Q33238). Agilent 4150 TapeStation and Genomic DNA Screen Tapes (Agilent Technologies, cat: 5067-5365) were used to measure DNA integrity. Sequencing libraries were prepared using Lotus DNA Library Prep Kit (IDT, cat: 10001074) and xGen Exome Research Panel v2 (IDT, cat: 10005153) for exon capture. After circularization using the MGIEasy Circularization Kit, enriched exome libraries were sequenced on DNBSEQ-G400RS using a standard 150 bp paired-end mode. Genomon v2.6.2 pipeline aligned sequencing reads to the human genome reference (GRCh37), and called the somatic variants with a mapping quality score ≥ 20; base quality score ≥15; exonic/splice-site mutations; strand ratio not equal to 0/1; depths in both tumor and normal ≥ 8; number of variant reads in the tumor ≥ 4; VAF ≥ 0.05 in tumor samples and <0.05 in normal samples; EBCall[60], $P \leq 10^{-3}$; and Fisher's exact test, $P \leq 10^{-1}$. Furthermore, visual inspection on the Integrative Genomics Viewer browser (https://software.broadinstitute.org/software/igv/) removed mapping errors. The supercomputer SHIROKANE was used for all analyses.

### Quantifying the *KMT2A* transcripts containing the intron 7 sequence

The amount of *KMT2A* transcripts was assessed by quantitative PCR (qPCR) using Power SYBR Green Master Mix (Thermo Fisher Scientific, cat: 4367659) and the StepOnePlus real-time PCR system (Thermo Fisher Scientific, cat: StepOnePlus-01). The sequences of the primers are as follows: 5′-TCAGAGTGCGAAGTCCCACAAG-3′ (exon 2 forward), 5′-GATTTTTACTCCAGGGAAGGTGG-3′ (exon 3 reverse), 5′-GGATGCCTTCCAAAGCCTACCTGC-3′ (exon 7 forward), 5′-CAATTCCTTCTTGCCCTCCTCAC-3′ (intron 7 reverse). Expression of exon 2 to exon 3 of the *KMT2A* transcript was used as a reference, and set as 1 for each sample. All measurements were carried out in duplicate, and the difference in the duplicate threshold cycles was less than one cycle for all samples analyzed. All experiments were repeated at least three times.

### Identification of the *KMT2A::AFF1* genomic junction

Before conducting the PCR to detect *KMT2A::AFF1*, M-MDSC-like AML cells were sorted from samples LS AML1, 2, 3, and 4. Thawed mononuclear cells from each sample were suspended in 100 μl PBS and stained at RT for 30 min with an-anti human CD45-PerCP/Cy5.5 (1:100, BioLegend, cat: 304028), an anti-human CD15-BV421 antibody (1:100, BioLegend, cat: 323040), an anti-mouse/human CD11b-FITC antibody (2:100, BioLegend, cat: 101206), an anti- human CD14-APC antibody (4:100, BioLegend, cat: 301808), an anti- human HLA-DR-PE antibody (1:100, BioLegend, cat: 307606), and Fixable Viability Dye eFluor 780 (1:2000, eBioscience, cat: 65-0865-14). After washing twice, each sample was diluted to $5 \times 10^6$ cells/ml with RPMI, and the M-MDSC-like AML (CD45[dim]CD15[-]CD11b[+]CD14[+]HLA-DR[low/-]) fraction was sorted using a FACSAria II (BD). The sorted AML cells were frozen in CELLBANKER 1 until required.

The genomic junctions of *KMT2A* and *AFF1* were identified by long distance PCR[61], but with some modifications. Briefly, 19 forward primers were designed to span every 300–700 base-pairs; this allowed coverage of the whole breakpoint cluster region of *KMT2A*. A mixture of all of these primers was used for PCR. As for *AFF1*, 48 reverse primers were designed to cover every 700 bps to 1.5 kb across exon 3 to exon 7 of *AFF1*; these primers were separated into four sets (A to D) for PCR amplification. PCR amplification of *KMT2A::AFF1* genomic sequence was performed in four separated tubes, each containing one of the *AFF1* reverse primer sets and all 19 *KMT2A* forward primers. Genomic DNA was extracted from peripheral blood using a standard procedure, and 10–50 ng of genomic DNA was subjected to PCR amplification. Long-distance PCR experiments were performed with Tks Gflex DNA Polymerase (Takara, cat: R060A). The PCR amplification conditions were as follows: 94 °C for 30 s, followed by 35 cycles of 94 °C for 10 s and 68 °C for 1 min. The second-round PCR was performed with a 1 μl aliquot of the first amplification product, and using internally nested primers. PCR products were electrophoresed in 1% agarose gels. Positive products were purified with a QIAquick Gel Extraction kit (Qiagen, cat: 28704) and sequenced directly on the SeqStudio Genetic

Analyzer (Applied Biosystems, cat: A35644) to identify the *KMT2A::AFF1* genomic junctions. An appropriate primer pair for each case was designed to span the identified genomic junction, and was then used for PCR to detect the *KMT2A::AFF1* fusion. Details of the primer sequences are available on request.

## Mass cytometry

Mononuclear cells were resuspended in RPMI 1640 medium (Sigma-Aldrich, cat: R8758) supplemented with 10% fetal calf serum and penicillin-streptomycin (Fujifilm Wako, cat: 168-23191), and then washed twice with phosphate buffered saline (PBS). Blocking was performed using Fc receptor Binding Inhibitor Polyclonal Antibody, Functional Grade (eBioscience, cat: 16-9161-73). Anti-chemokine antibodies were added to yield 100 µl final reaction volumes in RPMI, and samples were incubated at 37 °C for 45 min, washed twice with cell-staining medium (CSM; PBS containing 0.1% bovine serum albumin, 2 mM ethylenediaminetetraacetic acid, and 0.01% sodium azide), and then stained at room temperature (RT) for 45 min in 100 µl CSM containing anti-cell surface marker antibodies. Next, the cells were stained with PBS containing 500 nM dichloro-(ethylenediamine) palladium (II) (Sigma-Aldrich, cat: 574902-1 G) to assess viability. Intracellular staining of Foxp3 and CTLA-4 was conducted using the Foxp3/Transcription Factor Staining Buffer Set (Thermo Fisher Scientific, cat: 00-5523-00). Finally, 150 µl of PBS containing 2% formaldehyde (diluted from Pierce 16% Formaldehyde, Methanol-free, Thermo Fisher Scientific, cat: 28906) and 1:500 Cell-ID Intercalator-Rh (Standard BioTools, cat: 201103 A) was added, and the samples were stored at 4 °C overnight. Before a measurement, samples were washed once with CSM and twice with Cell Acquisition Solution (Standard BioTools, cat: 201240), filtered to remove aggregates, and resuspended in Cell Acquisition Solution containing 15% EQ Four Element Calibration Beads (Standard BioTools, cat: 201078). Throughout the analysis, cells were introduced at a constant rate of approximately 100–400 cells/s. The CyTOF antibody panel is presented in Supplementary Table 5.

## Generation of M-MDSCs from human PBMCs

As a positive control for the subsequent T cell suppression and Treg co-culture assays, M-MDSCs were generated from healthy PBMCs as described by a previous report that introduced the induction protocol[36]. Briefly, PBMCs were isolated by density centrifugation and plated at $1 \times 10^6$ cells/ml in RPMI 1640 medium supplemented with IL-6 (20 ng/ml; PEPROTECH, cat: AF-200-06-20ug), GM-CSF (20 ng/ml; R&D systems, cat: 215-GM), isoproterenol (Tokyo Chemical Industry, cat: I0260), 10% fetal calf serum, and penicillin-streptomycin. Cells were cultured for 6 days in a 37 °C/5% $CO_2$ incubator. The medium was replenished every 2–3 days. On Day 6, the induced M-MDSCs were sorted from the cultured PBMCs by magnetic sorting on CD14 MicroBeads (Miltenyi Biotec, cat: 130-050-201) and the autoMACS Pro Separator (Miltenyi Biotec, cat: 130-092-545). The yield and purity of the sorted M-MDSCs were confirmed by flow cytometry using a FACSLyric (BD) after staining at RT for 30 min with an anti-human CD15-BV421 antibody (1:100), an anti-mouse/human CD11b-FITC antibody (2:100), an anti-human CD14-APC antibody (4:100), an anti-human HLA-DR-PE antibody (1:100), and Fixable Viability Dye eFluor 780 (1:2000). The purity of the CD14+ cells was > 90%.

## T cell suppression assay

The T cell suppression assay of AML cells was performed in a manner similar to the human polyclonal suppression assay introduced in the previous experimental guidelines[62]. First, Naïve Tconv (CD4+CD25-CD45RA+ T cells) were obtained by cell sorting. Fresh PBMCs were stained at RT for 30 min with an anti-human CD4-APC antibody (1:100, BioLegend, cat: 300514), an anti-human CD25-BV421 antibody (2:100, BD, cat: 564033) and an anti-human CD45RA-PerCP/Cy5.5 antibody (1:100, BioLegend, cat: 304122). Then, Naïve

Tconv were sorted using a FACSAria II (BD), and Naïve Tconv were frozen in liquid nitrogen until required. When conducting the T cell suppression assay, thawed Naïve Tconv were stained at 37 °C for 20 min with 1 µM Cell Trace CFSE Cell Proliferation Kit reagent (Thermo Fisher Scientific, cat: C34570). In this study, Tregs were substituted with AML cells or induced M-MDSCs from three healthy donors. CFSE-stained Naïve Tconv ($1 \times 10^4$ cells) were cultured with AML cells or induced M-MDSCs at various ratios (1:0, 1:1, 1:2 [T cells:AML cells or induced M-MDSCs]) and then stimulated with Dynabeads Human T-activator CD3/CD28 (Thermo Fisher Scientific, cat: 11161D; bead-to-T cell ratio 1:1) in RPMI medium containing 10% fetal bovine serum, L-glutamine, IL-2 (Imunace 35, KYOWA Pharmaceutical Industry, cat: 876399) (30 IU/ml), and penicillin/streptomycin (final volume, 200 µl in a 96-well flat bottom plate. On Day 4 (12 h before flow cytometry analysis), brefeldin A (BioLegend, cat: 420601) was added to each well to a final concentration of 5 µg/ml. Five days after the start of co-culture, cells were stained at RT for 30 min with an anti-human CD4-APC antibody (1:100) and Fixable Viability Dye eFluor 780 (1:1000). The samples were then washed twice in PBS and stained intracellularly with an anti-human IFN-γ-BV421 antibody (5:100, BioLegend, cat: 506538) and the Foxp3/Transcription Factor Staining Buffer Set at 4 °C for 30 min. Data were collected on a flow cytometer (FACSLyric). Acquired data were analyzed by FlowJo software (v10.8.1) (BD Biosciences) and GraphPad Prism 9.5.1 (GraphPad Software).

## Treg co-culture assay

CD4+ T cells were sorted from fresh PBMCs obtained from a healthy donor using CD4+ T cell Isolation Kit (Miltenyi Biotec, cat: 130-096-533) and the autoMACS Pro Separator. Sorted CD4+ T cells were frozen in liquid nitrogen until required.

After thawing, $1 \times 10^4$ CD4+ T cells were cultured with AML cells or induced M-MDSCs (1:1 ratio) in 200 µl of RPMI medium supplemented with 10% fetal bovine serum without any stimulation in the 96-well round bottom plate. Three different induced M-MDSCs, generated from PBMCs from each healthy person, were used for the biological replicates. After 5 days, the co-cultured cells were washed twice with PBS. After blocking with Fc receptor Binding Inhibitor, fluorescent-labeled antibodies and Fixable Viability Dye eFluor 780 in PBS were added to yield a final reaction volume of 100 µl, and the samples were incubated at RT for 30 min. Samples were then washed twice with PBS and intracellular staining was conducted using the FOXP3/Transcription Factor Staining Buffer Set and an anti-Foxp3 antibody. Before flow cytometry analysis, samples were resuspended in 400 µl of CSM and measured using a FACSLyric cytometer. Acquired data were analyzed by Cytobank software (https://premium.cytobank.org/cytobank/login). The antibody panel used for flow cytometry is presented in Supplementary Table 8.

## In vitro drug sensitivity testing

To evaluate the drug sensitivity of AML cells from LS AML and LC AML, DST was designed in accordance with a previous report[37]. Briefly, cryopreserved AML cells were thawed and seeded at $1 \times 10^4$ live cells/10 µl onto a 384-well plate preloaded with 10 µl of RPMI 1640 medium supplemented with 20% of fetal calf serum containing one of 30 drugs at four serially diluted concentrations ($\times 1$, $\times 5^{-1}$, $\times 5^{-2}$, $\times 5^{-3}$). The drugs, and final concentrations used in the DST are listed in Supplementary Table 6. After a 96-h incubation at 37 °C in a fully humidified 5% $CO_2$ atmosphere, the viability of the cells in each well was measured in the CellTiter-Glo Luminescent Cell Viability Assay (Promega, cat: G7570), a chemiluminescent assay which detects live cells. To compare drug sensitivities between samples, the drug effect score (DES) was calculated as follows: DES = [(100 - % live cells at 1:125 dilution) × ln(125) + (100 - % live cells at 1:25 dilution) × ln(25) + (100 - % live cells at 1:5 dilution) × ln(5) + (100 - % live cells and undiluted)]/

(ln(125) + ln(25) + ln(5) + 1). A DES of 100 indicates that the drug killed all of the cells at every tested concentration, whereas a DES of 0 indicates that the drug had no effect.

If the blast ratio in a patient sample was not high, the DES was corrected according to the percentage of blasts as follows: corrected DES = [DES of a whole sample - (100 - % of blasts) × 0.01 × reference DES of normal blood cells]/ (% of blasts × 0.01).

## Statistical analyses and reproducibility

LS AML is a rare disease; therefore, samples are difficult to come by. Therefore, no statistical method could be used to predetermine sample size. The experiments were not randomized, and investigators were not blinded to the results. Statistical values in GSEA were calculated using GSEA software v 4.3.2[16]. DEGs between LS AML and LC AML were compared based on the DESeq2-derived -$\log_{10}$ adjusted $P$ values, with the signs of $\log_2$-fold changes. The proportion of M-MDSC-like blasts and Tregs in clinical samples from two groups was evaluated by a two-tailed Mann-Whitney test. As a rule, all in vitro experiments were repeated three times, with at least two biological replicates, and statistical analyses were performed using GraphPad Prism 9.5.1 (GraphPad Software). Comparison of more than two groups was made using one-way analysis of variance (ANOVA) with Turkey's multiple comparison test. For the T cell suppression assay, the difference in the proliferation index was tested by two-way ANOVA with Sidak's multiple comparisons test. Error bars and significant values are presented within the figure legends; *$p < 0.05$, **$p < 0.01$, ***$p < 0.001$, and ****$p < 0.0001$. No data were excluded from the analysis. In the in vitro experiments, data distribution was assumed to be normal, but this was not formally tested.

## Ethical regulation

This study was approved by the Kyoto University Hospital Ethical Board (G-1030, R-2831) and related institutions. The patients or their guardians provided informed consent for sample collection.

## Reporting summary

Further information on research design is available in the Nature Portfolio Reporting Summary linked to this article.

## Data availability

The whole exome sequencing data and RNA sequencing data for LS AML and LC AML (*KMT2A-r* AML), obtained from the in-house cohort, have been deposited in the Japanese Genome-phenotype Archive (JGA) managed by the DNA Data Bank of Japan (DDBJ) (https://www.ddbj.nig.ac.jp/index-e.html) under the JGAS000631[https://humandbs.dbcls.jp/en/hum0405-v2]. The instruction for accessing the data is indicated in [https://humandbs.dbcls.jp/en/data-use]. The results published herein are in part based upon data generated by the Therapeutically Applicable Research to Generate Effective Treatments (TARGET) (https://www.cancer.gov/ccg/research/genome-sequencing/target) initiative, phs000218. The data used for analysis are available at the Genomic Data Commons (https://portal.gdc.cancer.gov). The mass cytometry data have been deposited in Zenodo under the 15909980). All other data and materials supporting the results or analyses presented in this study are available upon reasonable request.

Software used in this study is open-source, except for Cytobank and FlowJo (both of which require payment prior to use). Details of the software versions and parameters are specified in the relevant subsections of the Methods.

All data are included in the Supplementary Information or available from the authors, as are unique reagents used in this Article. The raw numbers for charts and graphs are available in the Source Data file whenever possible. Source data are provided with this paper.

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

## Acknowledgements

This investigation was supported by the Japan Society of Promotion of Science (JSPS) KAKENHI (nos. JP20H00528, JP21K19405, and JP23K18264 to J.T.; and no. JP22K07211 to I.K.); by Project for Promotion of Cancer Research and Therapeutic Evolution (P-PROMOTE) grants from the Japan Agency for Medical Research and Development (AMED)

(nos. JP22cm0106xxxh000x and 23ama221505h0002 to J.T.; and nos. 22ama221514h0001 and 23ama221514h002 to M.T. and I.K.); by Practical Research for Innovative Cancer Control grants from AMED (no. JP21ck0106531 to I.K.; and no. JP24ck0106935 to Takako Miyamura); by the Princess Takamatsu Cancer Research Fund (to J.T.); by Takeda Hosho Grants for Research in Medicine (to J.T.); by the internal grant Ishizue from Kyoto University Research Administration to J.T.; Bristol Myers Squibb (to I.K.); by the Mochida Memorial Foundation for Medical and Pharmaceutical Research (to I.K.); by the Mother and Child Health Foundation (to I.K.); and by the Japan Leukemia Research Fund (to I.K.).

## Author contributions

Takashi Mikami, I.K., J.B.W., A.N., and M.T. conceived the project and provided leadership. S.H., S.T., S.I., T.H., K.I., H.T., and Takako Miyamura collected samples and contributed to scientific discussion. Takashi Mikami, A.N., T.K., K.T., H.K., Y.U., Y.N., and M.E.I. performed genomic analysis. Takashi Mikami and J.B.W. conducted CyTOF analysis and in vitro culture assays. H.G. conducted the drug sensitivity test. Tomoya Isobe, D.T., Toshihiko Imamura, S.O., M.T., M.E., and J.T. interpreted the data and contributed to scientific discussion. Takashi Mikami, A.N., and I.K. wrote the paper. All authors read the paper and agreed to the content.

## Competing interests

The authors declare no competing interests.

## Additional information

[1]Department of Pediatrics, Graduate School of Medicine, Kyoto University, Kyoto, Japan. [2]Department of Pediatrics and Developmental Biology, Institute of Science Tokyo, Tokyo, Japan. [3]Department of Pediatrics, Ehime University Graduate School of Medicine, Ehime, Japan. [4]Department of Pediatrics, Kurashiki Central Hospital, Okayama, Japan. [5]Wellcome-MRC Cambridge Stem Cell Institute, Department of Hematology, University of Cambridge, Cambridge, UK. [6]Department of Child Health and Welfare, Graduate School of Medicine, University of the Ryukyus, Okinawa, Japan. [7]Department of Pediatrics, Yokohama City University Graduate School of Medicine, Yokohama, Japan. [8]Department of Pediatrics, Kobe University Graduate School of Medicine, Kobe, Japan. [9]Department of Hematology and Oncology, Kobe Children's Hospital, Kobe, Japan. [10]Department of Pediatrics, the University of Osaka Graduate School of Medicine, Osaka, Japan. [11]Division of Leukemia and Lymphoma, Children's Cancer Center, National Center for Child Health and Development, Tokyo, Japan. [12]Department of Pediatrics, Kyoto Prefectural University of Medicine, Kyoto, Japan. [13]Department of Pediatrics, Mie University Graduate School of Medicine, Mie, Japan. [14]Division of Hematology/Oncology, Kanagawa Children's Medical Center, Yokohama, Japan. [15]Department of Pathology and Tumor Biology, Graduate School of Medicine, Kyoto University, Kyoto, Japan. [16]Institute for the Advanced Study of Human Biology (WPI-ASHBi), Kyoto, Japan. [17]Department of Medicine, Center for Hematology and Regenerative Medicine, Karolinska Institute, Stockholm, Sweden. [18]Laboratory of Human Single Cell Immunology, Immunology Frontier Research Center (IFReC), Osaka University, Osaka, Japan. [19]Human Single Cell Immunology Team, Center for Infectious Disease Education and Research (CiDER), Osaka University, Osaka, Japan. [20]Center for Advanced Modalities and DDS (CAMaD), Osaka University, Osaka, Japan. [21]These authors contributed equally: Itaru Kato, Junko Takita. [22]These authors jointly supervised this work: James Badger Wing, Junko Takita. ✉e-mail: itarkt@kuhp.kyoto-u.ac.jp; jtakita@kuhp.kyoto-u.ac.jp

