## [Transparent Peer Review file · Nature Communications]

Multi-omics analysis identifies an M-MDSC-like immunosuppressive phenotype in lineage-switched AML with KMT2A rearrangement

Corresponding Author: Dr Itaru Kato

Version 0:

Reviewer comments:

Reviewer #1

(Remarks to the Author)

Mikami et al aim to study the mechanisms of immune evasion in lineage switching (LS) MLLr-AML after CART or bispecific antibody treatment. The authors included 3 such patients and a cohort of lineage consistent (LC) MLLr-AML as comparison. To increase the case number of LS patients, they included 3 more cases reported in literature. Through bulk RNA-seq analysis, the LS cases cannot be separated from LC MLLr-AML. However, gene expression clustering identified 3 clusters with all 6 LS cases in cluster 2. Within this cluster, the authors compared LS to LC and showed LS had downregulation of immune responses, reminiscent of monocytic myeloid derived suppressor cells (M-MDSC). Next, they performed cytoF on 3 LS and 5 LC cases and showed increased M-MDSCs in LS. The authors then performed sc-RNA seq to demonstrate M-MDSCs populations at single cell level in LS. Lastly, they performed functional analysis showing these cells indeed suppressed T-cell proliferation and IFN γ production and increased Treg differentiation in vitro. While the phenotypic data of M-MDSCs appears interesting, the cohort is very small, the studies are largely descriptive, and the functional analysis is weak. Furthermore, it is unclear if these M-MDSCs share the genetic aberrations with AML blasts.

Major concerns:

1. The cohort is small with only 3 cases. Although 3 additional cases from a published study are included, making an observation/conclusion based on 6 cases may be biased. For example, LS1 had a low M-MDSCs (Fig S3), at a level similar to LC. Most difference was driven by LS2 and 3. The authors are encouraged to increase the size of the cohort by collaborating with others. Alternatively, there should be more such cases with RNA seq data available in the literature. If batch effects can be reasonably corrected, which is another concern, combining such cases may strengthen the observation.
2. The phenotypic definition is not consistently stated, even between flow cytometry and cytoF. It is unclear how the 5 LC cases were chosen for these comparisons.
3. The genetics of M-MDSCs should be illustrated by either calling fusion based on single cell RNA seq data or by evaluating sorted M-MDSCs.
4. The functional analysis relied on data from one patient, which is not convincing. It appears that the coculture was done between AML blasts and T cells, therefore, confounding factors including non M-MDSCs cannot be ruled out. When IFN γ was measured, it was from supernatant. A more convincing way is to do intracellular staining of T cells and show if IFN γ from T cells is reduced. The FOXP3 stain and gating strategy appear arbitrary without positive controls.
5. The authors did not provide mechanistic insights why LS cases have increased M-MDSCs. Is it related to the phenotypic switch? If that's the case, do you expect to see this in other LS cases? Or if it is related to particular genetic aberrations, not much to LS?

Reviewer #2

(Remarks to the Author)

The study aims to elucidate the genetic and biological mechanisms underlying immune evasion by LS leukemia cells, an intriguing and largely unexplored clinical issue. The authors employed multi-omics analyses to characterize LS AML cells with the KMT2A::AFF1 fusion, discovering that these cells exhibit features similar to monocytic myeloid-derived suppressor cells (M-MDSCs). In vitro assays using one pair of samples (one from LC and one from LS) demonstrated that LS AML cells have immunosuppressive capacities akin to MDSCs. The authors concluded that their study provides insights into the immunological mechanisms of LS AML, highlighting its MDSC-like properties.

The manuscript is well-written, with clear presentation of ideas. The experiments are well-designed, and the results appear reliable. However, I have one major concern regarding the MDSC properties in LS AML presented in this study. While the authors claim that MDSCs play a significant role in AML and are associated with worse survival from literature review, the relevance of MDSCs in AML is not as pronounced as in other hematologic malignancies, such as MDS. Although there are references, the evidence is still not compelling enough, in my opinion. As we know, the hallmark characteristic of MDSCs is their T-cell immunosuppressive function. Therefore, to substantiate the MDSC characteristics, in vitro studies of T-cell suppression are essential. In this study, the authors demonstrated in vivo assays but only with one pair of patients. Given the heterogeneity of AML and its numerous genetic dysregulations, one pair of samples seems insufficient to draw definitive conclusions about MDSC properties. However, considering the difficulty in obtaining samples and the lack of replaceable cell lines, this represents a major limitation. Aside from this issue, I have no further questions.

Reviewer #3

(Remarks to the Author)

In this manuscript the Authors, perform a multi-omics analysis of three samples from patients with lineage switching acute myeloid leukemia, and identify that the lineage switched AML population exhibit a similar phenotype to monocytic myeloid-derived suppressor cells. Furthermore, in vitro suppression assays confirmed the suppressive function of those cells. The findings are timely and important, if we consider the poor clinical outcome of those patients. However, some concerns were raised that may assist to strengthen the findings of the manuscript.

1. In page 9 the finding of low HLADR expression used to conclude that LS AML cells may have an affinity with M-MDSCs. I found this connection weak, since other cell of myeloid lineage may downregulate HLA molecules due to immunosuppressive microenvironment (i.e high IL-10 content).
2. In page 10, they refer to "non-tumor" MDSCs, but to my understanding these are MDSCs from the patients, and therefore are tumor MDSCs but not the LS AML-related. It would be of interest to include data from blood from healthy individuals (which may have low M-MDSCs) or from another hematologic malignancy.
3. In a transcriptomic level, what are the differences between LS AML-like M-MDSC and the M-MDSCs? Could the authors also show the differences in CD45 expression between the two cell populations?
4. I am puzzled with the in vitro suppression assay, and the rationale for using healthy donor PBMCs. Is this like a mixed leukocyte reaction? They isolated CD3+ (which include the CD8+ t cells) and may recognize MHC I molecules on the AML cells to cause a rapid proliferation. Also, important controls are missing like the "normal M-MDSCs). Finally, the authors claim for a Treg cell induction upon co-culture of PBMCs with LS AML cells. However it could be just Treg cell proliferation (or increased Foxp3 MFI) since Treg cells preexist in the PBMC sample (also evident in Fig5f).

Reviewer #4

(Remarks to the Author)

Mikami et. al. investigate the very important clinical problem of CD19+ B cell acute lymphoblastic leukemia recurring as acute myelogenous leukemia after immunotherapy (LS AML). They present interesting observations that LS AML samples display genetic signatures, protein expression, and function similar to monocytic MDSC via gene set enrichment analysis, other NGS and omics analyses, CyToF, flow cytometry, and suppression assays. Overall the methodology is solid and the data are intriguing, but in the end enough samples are not analyzed to be able to draw any sound conclusions about the results and as a result the study does not significantly extend the current literature. The following deficiencies should be addressed:

Comment: Pg 8 line 21 Did lower MHC II in LS-AML come from lower MHC II expression in the CD19+ ALL? i.e. what was the MHC II expression of the primary ALL? Since LC AML have high MHC II, could low MHC II be a marker for ALL that will relapse as LS-AML under targeted immunotherapy? This may be what fig 2d shows (line 17 pg 9) or especially fig 3 – but the sample identities are somewhat unclear.

Lines 10-11 mention using normal M-MDSC from a healthy donor. However, MDSC are defined as pathologically activated cells [Veglia F, Sanseviero E, Gabrilovich DI. Myeloid-derived suppressor cells in the era of increasing myeloid cell diversity. *Nat Rev Immunol.* 2021 Aug;21(8):485-498. doi: 10.1038/s41577-020-00490-y. Epub 2021 Feb 1. PMID: 33526920; PMCID: PMC7849958]. MDSC by definition do not exist in a healthy person. Gabrilovich DI. Myeloid-Derived Suppressor Cells. *Cancer Immunol Res.* 2017 Jan;5(1):3-8. doi: 10.1158/2326-6066.CIR-16-0297. PMID: 28052991; PMCID: PMC5426480. While these "normal" MDSC were taken from the BM of a healthy individual where around 20% of myeloid cells will have markers similar to M-MDSC, it is unclear what these cells would be and are likely a transient immature cell in normal development. It is unclear what comparison of these cells with LS AML and LC AML really means. Maybe better to compare with MDSC from benign acute or chronic infected patients? Also, only 1 normal donor was used.

Fig 5d. MDSC are also reported to make IFN γ , so ELISA analysis cannot attributed IFN γ production to T cells. Should also include intracellular flow cytometry to assess T cell IFN γ directly. Or perhaps LS AML2 cells bind up IFN γ at a higher rate than LC AML? This figure being based on n=1 samples is not appropriate at all. This could easily be due to patient or sample variation. 5b difference of 2 proliferation peaks could easily be LS/LC patient sample differences of metabolism, MHC I expression, MHC haplotype matching, etc. and does not necessarily indicate there is any immunosuppression.

Supplemental fig 5, gemcitabine and docetaxel are not specific for MDSC and are broadly active on many cells. With the small number of samples, this could easily be attributed to sample variation. As a result, these data are not informative.

It is pointed out by the authors, and very understandable that this condition is rare and samples are hard to get. However, there is not nearly enough power for statistical significance, and statistically, nothing can be concluded from the study. As a result, this report does not really add to what is already known, i.e. there are other publications indicating AML cells are immunoinhibitory cells appearing M-MDSC-like in several ways, and since B-ALL has been shown to be immunosuppressive, [Hunter, R., Imbach, K.J., Zhou, C. et al. B-cell acute lymphoblastic leukemia promotes an immune suppressive microenvironment that can be overcome by IL-12. *Sci Rep* 12, 11870 (2022). <https://doi.org/10.1038/s41598-022-16152-z>] it may be that the MDSC-like appearance of some of the LS-AML cells is coincident rather than a cause of the immunosuppression in LS AML.

This paper presents some very interesting observations, but provides only underpowered correlative data. The sample numbers are just too small. This report is perhaps suggestive, but not definitive in any way.

Version 1:

Reviewer comments:

Reviewer #1

(Remarks to the Author)

I applaud to the authors for the efforts they spent addressing the concerns I raised in the initial review. They have done a great job and answered most points. Here are a few additional suggestions:

1. p8 line 6-7: one of the potential explanations that LS AML is in cluster 2 is due to selection bias as they harbor AFF1 fusion. The authors should at least mention it.
2. p8 line 21-23, related to the above concerns, the authors should compare LS AML with other clusters (not only LC AML in cluster2) to see if similar conclusion can be drawn.
3. p10, line 9-14, regarding these variants identified by WES, I have major concerns. first, VAF of many variants are very high including KRAS (KRAS or NRAS are typically subclonal with low VAF). second, TP53 variants seem extremely common, which is contradictory to what we know in clinical patients that TP53 mutations are exceedingly rare in MLL-r patients. Therefore, I highly suspect many of these variants are SNPs rather than pathogenic mutations. The authors should re-examine the data and validate it seriously. Related to this, the whole discussion regarding TP53 needs to be removed.
4. page 11, line 14-15, the statement of non-AML derived M-MDSCs is questionable. First, only one case showed a few cells in monocytes; second, there is no genetic evidence showing these cells do not carry MLL fusion. So the author should instead emphasize that M-MDSCs are largely leukemic cells. Relatedly, line 22-23 should be removed.

Reviewer #2

(Remarks to the Author)

All of my questions and concerns have been addressed. I have no further comments, and I believe the manuscript is suitable for publication.

Reviewer #3

(Remarks to the Author)

The authors addressed successfully my comments

Reviewer #4

(Remarks to the Author)

The authors have sufficiently addressed my major concern of insufficient sample numbers by obtaining more independent samples that strengthened their observations. I am comfortable with their conclusions now. Also, they have sufficiently addressed my other concerns of MHC II expression, proper MDSC controls, attributing IFN γ expression directly to T cells, and addressing caveats where appropriate.

Responses to comments made by the Reviewers

Dear Reviewers,

We are sincerely grateful to you for your interest in our research, and for the positive and encouraging feedback. We fully acknowledge the valid concerns regarding the limited number of cases in our original cohort. As pointed out, lineage-switched AML is extremely rare; nevertheless, we made every effort to address this by reaching out to pediatricians and hematologists across Japan, and were fortunate enough to obtain two additional lineage-switched samples. Furthermore, we identified and incorporated one more publicly available RNA-seq dataset of LS AML. As a result, we were able to expand our cohort to nine LS AML cases, which allowed us to confirm the reproducibility of our data, and increase the robustness and generalizability of our findings. In addition, we revised the experimental design of the *in vitro* assays by increasing the number of biological replicates and including induced M-MDSCs as a positive control. In the revised manuscript, the expanded bulk RNA-seq analysis including the additional LS samples further clarified the M-MDSC-like properties of LS AML. In addition to lower expression of MHC class II genes, we observed downregulation of gene sets related to the acquired immune response.

Our detailed point-by-point responses to each of the reviewers' suggestions and comments, and explaining our additional analyses and experiments, are set out below. All revisions are highlighted in red. Again, we greatly appreciate your kind guidance and thoughtful correspondence. Please do not hesitate to contact us again should you require further assistance/clarification.

Reviewer comments

Reviewer #1 (Remarks to the Author):

Mikami et al aim to study the mechanisms of immune evasion in lineage switching (LS) MLLr-AML after CART or bispecific antibody treatment. The authors included 3 such patients and a cohort of lineage consistent (LC) MLLr-AML as comparison. To increase the case number of LS patients, they included 3 more cases reported in literature. Through bulk RNA-seq analysis, the LS cases cannot be separated from LC MLLr-AML. However, gene expression clustering identified 3 clusters with all 6 LS cases in cluster 2. Within this cluster, the authors compared LS to LC and showed LS had downregulation of immune responses, reminiscent of monocytic myeloid derived suppressor cells (M-MDSC). Next, they performed cytoF on 3 LS and 5 LC cases and showed increased M-MDSCs in LS. The authors then performed sc-RNA seq to demonstrate M-MDSCs populations at single cell level in LS. Lastly, they performed functional analysis showing these cells indeed suppressed T-cell proliferation and IFN γ production and increased Treg differentiation *in vitro*. While the phenotypic data of M-MDSCs appears interesting, the cohort is very small, the studies are largely descriptive, and the functional analysis is weak. Furthermore, it is unclear if these M-MDSCs share the genetic aberrations with AML blasts.

Major concerns:

1. The cohort is small with only 3 cases. Although 3 additional cases from a published study are included, making an observation/conclusion based on 6 cases may be biased. For example, LS1 had a low M-MDSCs (Fig S3), at a level similar to LC. Most difference was driven by LS2 and 3. The authors are encouraged to increase the size of the cohort by collaborating with others. Alternatively, there should be more such cases with RNA seq data available in the literature. If batch effects can be reasonably corrected, which is another concern, combining such cases may strengthen the observation.

[Answer]

Thank you very much for your valuable and insightful comments. As you rightly point out, lineage switching is an extremely rare event in leukemia. Motivated by your important suggestion, we made every effort to collect additional cases, and were fortunate enough to obtain two more lineage-switched samples. In addition, we identified one more publicly deposited RNA-seq dataset of LS AML. Although information about FAB classification was not available for the deposited RNA-seq case, supplemental immunophenotyping data suggested a monocytic lineage; therefore, we included it in the analysis. Thus, we were able to expand our LS cohort to nine cases, thereby increasing the robustness of our findings.

2. The phenotypic definition is not consistently stated, even between flow cytometry and cytoF. It is unclear how the 5 LC cases were chosen for these comparisons.

[Answer]

We sincerely apologize for the lack of clarity regarding the phenotypic definition in the original manuscript. Consistent with previous studies, we defined leukemia cells as CD45^{neg-dim} myeloid lineage cells (CD33⁺ and/or CD13⁺)^{1,2}. This definition was also used to sort AML cells from patient samples in flow cytometry analysis (page 12, lines 13–16). Regarding selection of in-house LC cases, we comprehensively searched our institutional database and identified preserved samples from patients with *KMT2A*-rearranged monocytic AML. We were thus able to collect five LC cases for comparative analysis.

3. The genetics of M-MDSCs should be illustrated by either calling fusion based on single cell RNA seq data or by evaluating sorted M-MDSCs.

[Answer]

We are really grateful for this advice. We agree that it is important to verify whether the M-MDSC-like population harbors the *KMT2A* rearrangement. Regarding cases LS AML1, 2, 3, and 4, for which we had sufficient sample for sorting, we sorted M-MDSC-like AML cells and identified *KMT2A::AFF1* fusion by either RNA-seq or nested PCR. These results have been included in the revised manuscript (page 12, lines 11–23).

4. The functional analysis relied on data from one patient, which is not convincing. It appears that the coculture was done between AML blasts and T cells, therefore, confounding factors including non M-MDSCs cannot be ruled out.

When IFN γ was measured, it was from supernatant. A more convincing way is to do intracellular staining of T cells and show if IFN γ from T cells is reduced. The FOXP3 stain and gating strategy appear arbitrary without positive controls.

[Answer]

Thank you very much for your constructive suggestions regarding the *in vitro* functional assays. In response, we revised the experimental design and increased the number of biological replicates to three cases each for LC AML and LS AML. As you correctly point out, the co-culture assays involved total AML blasts and T cells, meaning that we cannot completely exclude confounding effects due to non-M-MDSC-like AML blasts. The practical reason for using total blasts was that the co-culture assay required 5 days of incubation, and cell sorting-induced stress compromises the viability of AML cells significantly. Moreover, the amount of available LS AML samples was limited, and sorting M-MDSC-like populations would have left us with too few cells for the assays. Nevertheless, our RNA-seq analysis showed that LS AML blasts as a whole exhibit more M-MDSC-like gene expression features than LC AML blasts, which supports the validity of this approach. These limitations have now been mentioned explicitly in the revised manuscript (page 20, lines 18–20). Regarding the method used to measure IFN- γ , we performed intracellular staining of T cells as suggested, and found that production of IFN- γ in T cells fell as co-culture time with LS AML cells increased. The result is now included in the revised manuscript (page 15, lines 10–23).

For the Treg co-culture assay, we used CD4⁺ T cells isolated from healthy PBMCs as a control to define effector Tregs (Fraction II) based on Foxp3/CD45RA gating (as mentioned in previous reports^{3,4}). Specifically, a perpendicular line was drawn on the right side of the Fraction I population; the CD45RA⁻Foxp3^{high} population to the right of this line was designated as Fraction II. An identical gating strategy was used to calculate the percentage of effector Tregs in all other co-cultured samples. This is now explained on page 16, lines 4–7, and in Figure 5f. Thank you again for your thoughtful guidance.

5. The authors did not provide mechanistic insights why LS cases have increased M-MDSCs. Is it related to the phenotypic switch? If that's the case, do you expect to see this in other LS cases? Or if it is related to particular genetic aberrations, not much to LS?

[Answer]

Thank you for raising this important question. In the context of immune escape, AML cells alter the immune microenvironment through mechanisms such as overexpression of inhibitory ligands, suppression of NK and T cell function, downregulation of HLA expression, and induction of MDSCs.

Although further investigations are warranted, our additional RNA-seq analysis performed during the revision process revealed that expression of *IRF4* was significantly lower in LS AML than in LC AML. Recently, IRF4 has been recognized as a critical regulator of hematopoietic progenitor cell fate decisions⁵, and its downregulation is implicated in promotion of MDSC development⁶.

While revising the paper, project EVOLVE (an international analysis of post-immunotherapy lineage switching)

was published⁷. This project evaluated clinical data from 75 LS cases and reported an increased frequency of LS following immunotherapy. Based on these observations, we speculate that immune pressure exerted during targeted immunotherapy may contribute to lineage switching as a form of immune escape, and that downregulation of *IRF4* may represent one of the factors that facilitates acquisition of an M-MDSC-like phenotype during this process. However, we would like to emphasize that this hypothesis remains speculative, and that further functional studies are required to establish a direct causal relationship between downregulation of *IRF4* and lineage switching. We have discussed this in the revised manuscript (page 18, lines 11–page 19, line 6).

Reviewer #2 (Remarks to the Author):

The study aims to elucidate the genetic and biological mechanisms underlying immune evasion by LS leukemia cells, an intriguing and largely unexplored clinical issue. The authors employed multi-omics analyses to characterize LS AML cells with the *KMT2A::AFF1* fusion, discovering that these cells exhibit features similar to monocytic myeloid-derived suppressor cells (M-MDSCs). *In vitro* assays using one pair of samples (one from LC and one from LS) demonstrated that LS AML cells have immunosuppressive capacities akin to MDSCs. The authors concluded that their study provides insights into the immunological mechanisms of LS AML, highlighting its MDSC-like properties.

The manuscript is well-written, with clear presentation of ideas. The experiments are well-designed, and the results appear reliable. However, I have one major concern regarding the MDSC properties in LS AML presented in this study. While the authors claim that MDSCs play a significant role in AML and are associated with worse survival from literature review, the relevance of MDSCs in AML is not as pronounced as in other hematologic malignancies, such as MDS. Although there are references, the evidence is still not compelling enough, in my opinion. As we know, the hallmark characteristic of MDSCs is their T-cell immunosuppressive function. Therefore, to substantiate the MDSC characteristics, *in vitro* studies of T-cell suppression are essential. In this study, the authors demonstrated *in vivo* assays but only with one pair of patients. Given the heterogeneity of AML and its numerous genetic dysregulations, one pair of samples seems insufficient to draw definitive conclusions about MDSC properties. However, considering the difficulty in obtaining samples and the lack of replaceable cell lines, this represents a major limitation. Aside from this issue, I have no further questions.

[Answer]

We would like to express our deep gratitude for your profound insight and thoughtful comments. It is true that the relevance of MDSCs in AML is not as well established as it is in other hematological diseases such as MDS. Although the precise reasons remain unclear, one possible explanation is that the acute expansion of blasts characteristic of AML, combined with intensive chemotherapy or HSCT, may simultaneously deplete other immune cells, including MDSCs. Nonetheless, research interest in the role of MDSCs in AML is increasing, and several clinical trials targeting MDSCs are ongoing⁸; the results of these trials should be followed carefully. We have now discussed this in the revised manuscript (page 17, lines 9–12). Fortunately, with the cooperation of pediatricians and hematologists across Japan, we were able to obtain additional LS AML samples and performed new *in vitro* assays comparing three cases each of LS AML and LC AML. The results were consistent with our previous findings, and have been incorporated

into the revised manuscript (page 15, line 10–page 16, line 11).

As you rightly point out, research on LS AML remains limited, and our study was inevitably constrained by the rarity of available samples. Nevertheless, careful investigation of these valuable specimens has allowed us to reinforce our initial findings. We believe that the steady accumulation of such studies, each providing incremental insights/advances, will ultimately lead to a deeper understanding of LS AML biology and, in turn, to new therapeutic strategies for this challenging disease.

Reviewer #3 (Remarks to the Author):

In this manuscript the Authors, perform a multi-omics analysis of three samples from patients with lineage switching acute myeloid leukemia, and identify that the lineage switched AML population exhibit a similar phenotype to monocytic myeloid-derived suppressor cells. Furthermore, in vitro suppression assays confirmed the suppressive function of those cells.

The findings are timely and important, if we consider the poor clinical outcome of those patients. However, some concerns were raised that may assist to strengthen the findings of the manuscript.

1. In page 9 the finding of low HLADR expression used to conclude that LS AML cells may have an affinity with M-MDSCs. I found this connection weak, since other cell of myeloid lineage may downregulate HLA molecules due to immunosuppressive microenvironment (i.e high IL-10 content).

[Answer]

We are sincerely grateful for your insightful comment, from which we have learned a great deal. We agree that low or negative expression of HLA-DR is observed in various myeloid-lineage cells under immunosuppressive conditions, and that this alone may not be sufficient to define a link to M-MDSCs. In our study, the observation of decreased HLA-DR expression by LS AML cells served initially as a conceptual entry point, prompting further investigation into their immunophenotypic and transcriptomic characteristics. To strengthen this hypothesis, we conducted GSEA using an established M-MDSC gene signature, which revealed significant enrichment of M-MDSC-related genes in LS AML. Furthermore, re-analysis of the expanded bulk RNA-seq dataset demonstrated a more pronounced downregulation of gene sets related to the acquired immune response in LS AML than in LC AML, reinforcing the notion of M-MDSC-like immunosuppressive properties. These results have been included in the revised manuscript (page 8, line 17–page 9, line 14; Fig. 2a–c).

2. In page 10, they refer to “non-tumor” MDSCs, but to my understanding these are MDSCs from the patients, and therefore are tumor MDSCs but not the LS AML-related. It would be of interest to include data from blood from healthy individuals (which may have low M-MDSCs) or from another hematologic malignancy.

[Answer]

Thank you very much for your insightful comment. As you correctly point out, the term “non-tumor” was inaccurate in this context, and has been revised to “non-AML-derived” in the revised manuscript (page 10, line 21). M-MDSCs

constitute approximately 0.5–3% of PBMCs in healthy individuals⁹⁻¹¹; the percentage in our healthy control PBMCs (used as a reference in the CyTOF analysis) was 2.3% (Supplementary Fig. 3a). We have added this information to the revised manuscript (page 11, lines 9–12).

3. In a transcriptomic level, what are the differences between LS AML-like M-MDSC and the M-MDSCs? Could the authors also show the differences in CD45 expression between the two cell populations?

[Answer]

Thank you very much for this important and insightful question. We agree that a transcriptomic comparison between M-MDSC-like LS AML cells and canonical M-MDSCs would be highly informative. During the single-cell RNA-seq analysis performed for the revision, we compared the gene expression pattern between M-MDSC-like LS AML cells and M-MDSCs from patients with autoimmune diseases. Interestingly, both had a similar immunological gene expression profile; however, unlike M-MDSCs the M-MDSC-like LS AML cells also expressed the neutrophil-myeloid progenitor signature. Considering that M-MDSC-like LS AML cells originate from AML, a kind of immature abnormal myeloid cell, it is not surprising that M-MDSC-like LS AML cells have such a neutrophil-myeloid progenitor signature. These results have been included in the revised manuscript (page 13, line 8–page 14, line 21; Fig. 4, and Supplementary Fig. 7). Regarding expression of CD45, we compared levels between M-MDSCs and M-MDSC-like AML cells using CyTOF analysis. The results showed no significant difference between the two populations (page 11, line 22–23; Supplementary Fig. 3b).

4. I am puzzled with the *in vitro* suppression assay, and the rationale for using healthy donor PBMCs. Is this like a mixed leukocyte reaction? They isolated CD3⁺ (which include the CD8⁺ t cells) and may recognize MHC I molecules on the AML cells to cause a rapid proliferation. Also, important controls are missing like the “normal M-MDSCs). Finally, the authors calin for a Treg cell induction upon co-culture of PBMCs with LS AML cells. However it could be just Treg cell proliferation (or increased Foxp3 MFI) since Treg cells preexist in the PBMc sample (also evident in Fig5f.

[Answer]

Thank you very much for your detailed and insightful comments regarding the *in vitro* suppression and Treg assays. As you correctly point out, our initial experimental design lacked important controls and clarity of interpretation. Based on your suggestions, we repeated the suppression assay using responder naïve Tconv cells (CD4⁺CD25⁻CD45RA⁺ T cells), and used induced M-MDSCs as a positive control. This refinement, prompted by your input, has improved the rigor of the assay and the reproducibility of our findings (page 15, lines 10–23; Fig. 5a–d).

Similarly, we repeated the Treg co-culture assay using responder CD4⁺ T cells. As you noted, the observed increase in effector Tregs could reflect proliferation of pre-existing Tregs within the CD4⁺ population. Your comment encouraged us to clarify our aim: to determine whether LS AML cells can increase the number of effector Tregs regardless of whether this occurs via induction or proliferation. This interpretation is consistent with prior studies reporting that MDSCs may contribute to both mechanisms¹². Accordingly, we have replaced the term “induce Tregs”

with “increase Tregs” in the revised manuscript (page 16, line 1–11). We sincerely appreciate your thoughtful suggestions, which have helped us to refine and improve the presentation of these experiments.

Reviewer #4 (Remarks to the Author):

Mikami et. al. investigate the very important clinical problem of CD19+ B cell acute lymphoblastic leukemia recurring as acute myelogenous leukemia after immunotherapy (LS AML). They present interesting observations that LS AML samples display genetic signatures, protein expression, and function similar to monocytic MDSC via gene set enrichment analysis, other NGS and omics analyses, CyToF, flow cytometry, and suppression assays. Overall the methodology is solid and the data are intriguing, but in the end enough samples are not analyzed to be able to draw any sound conclusions about the results and as a result the study does not significantly extend the current literature. The following deficiencies should be addressed:

Comment: Pg 8 line 21: Did lower MHC II in LS-AML come from lower MHC II expression in the CD19+ ALL? i.e. what was the MHC II expression of the primary ALL? Since LC AML have high MHC II, could low MHC II be a marker for ALL that will relapse as LS-AML under targeted immunotherapy? This may be what fig 2d shows (line 17 pg 9) or especially fig 3 – but the sample identities are somewhat unclear.

[Answer]

We sincerely thank you for your sharp and insightful observations. As you suggest, we investigated expression of HLA-DR in paired ALL and LS AML samples from the same patient. In every ALL sample, HLA-DR was highly expressed, with positivity exceeding 95%. By contrast, expression of HLA-DR fell markedly after lineage switching to AML (page 11, line 23–page 12, line 3; Supplementary Fig. 3c). These findings indicate that the observed reduction in expression of MHC class II in LS AML is unlikely to be inherited from the original ALL phenotype.

Lines 10-11 mention using normal M-MDSC from a healthy donor. However, MDSC are defined as pathologically activated cells [Veglia F, Sanseviero E, Gabrilovich DI. Myeloid-derived suppressor cells in the era of increasing myeloid cell diversity. Nat Rev Immunol. 2021 Aug;21(8):485-498. doi: 10.1038/s41577-020-00490-y. Epub 2021 Feb 1. PMID: 33526920; PMCID: PMC7849958]. MDSC by definition do not exist in a healthy person. Gabrilovich DI. Myeloid-Derived Suppressor Cells. Cancer Immunol Res. 2017 Jan;5(1):3-8. doi: 10.1158/2326-6066.CIR-16-0297. PMID: 28052991; PMCID: PMC5426480. While these "normal" MDSC were taken from the BM of a healthy individual where around 20% of myeloid cells will have markers similar to M-MDSC, it is unclear what these cells would be and are likely a transient immature cell in normal development. It is unclear what comparison of these cells with LS AML and LC AML really means. Maybe better to compare with MDSC from benign acute or chronic infected patients? Also, only 1 normal donor was used.

[Answer]

Thank you very much for your insightful comment, and for highlighting the important conceptual distinction in the definition of MDSCs. As noted in our response to Reviewer 3, phenotypically M-MDSC-like cells can be detected at

low frequencies (approximately 0.5–3%) in PBMCs from healthy individuals⁹⁻¹¹, and our CyTOF analysis of a healthy control confirmed a similar proportion (2.3%) (Supplementary Fig. 3a, and page 11, lines 9–12); however, we fully agree with your critical point that MDSCs, by definition, are pathologically activated (and do not exist under steady-state physiological conditions). In light of your comment, and to better capture this pathological context, we expanded our single-cell RNA-seq analysis to include bone marrow samples from three patients with chronic inflammatory diseases: systemic lupus erythematosus, juvenile idiopathic arthritis, and Sjögren’s syndrome. As a result, we found that M-MDSC-like LS AML cells and M-MDSCs from the patients with autoimmune diseases had similar immunological gene expression profiles; however, unlike M-MDSCs, M-MDSC-like LS AML cells were also characterized by a neutrophil-myeloid progenitor signature. The results have been included in the revised manuscript (page 13, line 8–page 14, line 21; Fig. 4, and Supplementary Fig. 7). Thank you for inspiring us to consider these similarities and differences.

Fig 5d. MDSC are also reported to make IFN γ , so ELISA analysis cannot attributed IFN γ production to T cells. Should also include intracellular flow cytometry to assess T cell IFN γ directly. Or perhaps LS AML2 cells bind up IFN γ at a higher rate than LC AML? This figure being based on n=1 samples is not appropriate at all. This could easily be due to patient or sample variation. 5b difference of 2 proliferation peaks could easily be LS/LC patient sample differences of metabolism, MHC I expression, MHC haplotype matching, etc. and does not necessarily indicate there is any immunosuppression.

[Answer]

We are deeply grateful for your thoughtful and scientifically important comments, which have helped us to improve the design and interpretation of our *in vitro* suppression assay significantly. As you correctly point out, IFN- γ measured by ELISA may not reliably reflect T cell-specific production, as MDSCs themselves can also produce IFN- γ or bind it through cell surface receptors. In response, we revised the experimental design by increasing the number of biological replicates (n = 3 independent LS AML cases) and performing intracellular cytokine staining to directly assess IFN- γ production by responder T cells. In addition, as per your suggestion, and consistent with published experimental guidelines¹³, we replaced CD3⁺ responder T cells with purified naïve Tconv cells to reduce variability and increase assay specificity. Under these refined experimental conditions, we were able to recapitulate the previous results with improved reproducibility and interpretability. These findings are described in the revised manuscript (page 15, lines 10–23; Fig. 5a–d).

Supplemental fig 5, gemcitabine and docetaxel are not specific for MDSC and are broadly active on many cells. With the small number of samples, this could easily be attributed to sample variation. As a result, these data are not informative.

[Answer]

Thank you very much for your constructive and important comment. As you rightly note, gemcitabine and docetaxel are not specific to MDSCs, and may affect a broad range of myeloid and non-myeloid cells. In addition, given the

limited number of LS and LC AML samples available, the observed differences may indeed be confounded by sample-to-sample variability. Accordingly, we have revised the interpretation of these results in the main text to present the findings descriptively, and with appropriate caveats (page 16, lines 17–21). Nevertheless, given the rarity of LS AML, we believe that compiling such exploratory data, even with acknowledged limitations, contributes to a cumulative understanding of this poorly characterized disease.

It is pointed out by the authors, and very understandable that this condition is rare and samples are hard to get. However, there is not nearly enough power for statistical significance, and statistically, nothing can be concluded from the study. As a result, this report does not really add to what is already known, i.e. there are other publications indicating AML cells are immunoinhibitory cells appearing M-MDSC-like in several ways, and since B-ALL has been shown to be immunosuppressive, [Hunter, R., Imbach, K.J., Zhou, C. et al. B-cell acute lymphoblastic leukemia promotes an immune suppressive microenvironment that can be overcome by IL-12. *Sci Rep* 12, 11870 (2022). <https://doi.org/10.1038/s41598-022-16152-z>] it may be that the MDSC-like appearance of some of the LS-AML cells is coincident rather than a cause of the immunosuppression in LS AML.

This paper presents some very interesting observations, but provides only underpowered correlative data. The sample numbers are just too small. This report is perhaps suggestive, but not definitive in any way.

[Answer]

We sincerely appreciate your thoughtful comments, and your understanding of the challenges associated with obtaining rare LS AML samples. As you rightly point out, rigorous interpretation requires careful consideration of cohort size and statistical power. In response, we have made every effort to collect and integrate additional data: we were able to obtain two additional LS AML cases with preserved fresh-frozen samples, as well as one deposited RNA-seq dataset. This allowed us to perform RNA-seq on a total of nine LS AML samples (including the five in-house cases), CyTOF on five cases, single-cell RNA-seq on five cases, and functional *in vitro* assays with three biological replicates. We are convinced that these additions to the the revised manuscript deepen our understanding of the pathophysiology of LS and its immunosuppressive characteristics. Most previous reports regarding LS are case reports¹⁴, and recent publications (Tirtakusuma et al., *Blood* (2022) and Bataller and Abuasab et al., *Haematologica* (2024) reported ten and six cohorts, respectively^{15,16}. Considering this current situation, the scale of our research is not necessarily too small, and we believe that it is precise due to the careful accumulation and analysis of rare and well-characterized LS AML samples. Our analyses allow us to approach the biological truth underlying this challenging condition and overcome LS in the future. We appreciate your understanding regarding this matter.

References

1. Ratei R., *et al.* Discriminant function analysis as decision support system for the diagnosis of acute leukemia with a minimal four color screening panel and multiparameter flow cytometry immunophenotyping. *Leukemia* **21**, 1204-1211 (2007).

2. Tsai A. G., *et al.* Multiplexed single-cell morphometry for hematopathology diagnostics. *Nat Med* **26**, 408-417 (2020).
3. Miyara M., *et al.* Functional delineation and differentiation dynamics of human CD4⁺ T cells expressing the FoxP3 transcription factor. *Immunity* **30**, 899-911 (2009).
4. Wing J. B., Tanaka A., Sakaguchi S. Human FOXP3(+) Regulatory T Cell Heterogeneity and Function in Autoimmunity and Cancer. *Immunity* **50**, 302-316 (2019).
5. Wang S., He Q., Ma D., Xue Y., Liu F. Irf4 Regulates the Choice between T Lymphoid-Primed Progenitor and Myeloid Lineage Fates during Embryogenesis. *Dev Cell* **34**, 621-631 (2015).
6. Nam S., *et al.* Interferon regulatory factor 4 (IRF4) controls myeloid-derived suppressor cell (MDSC) differentiation and function. *J Leukoc Biol* **100**, 1273-1284 (2016).
7. Silbert S. K., *et al.* Project Evolve, Evaluation of Lineage Switch (LS), an International Initiative: Preliminary Results Reveal Dismal Outcomes in Patients with LS. *Blood* **142**, 4202-4202 (2023).
8. Wang S., Zhao X., Wu S., Cui D., Xu Z. Myeloid-derived suppressor cells: key immunosuppressive regulators and therapeutic targets in hematological malignancies. *Biomark Res* **11**, 34 (2023).
9. Kotsakis A., Harasymczuk M., Schilling B., Georgoulas V., Argiris A., Whiteside T. L. Myeloid-derived suppressor cell measurements in fresh and cryopreserved blood samples. *J Immunol Methods* **381**, 14-22 (2012).
10. Apodaca M. C., *et al.* Characterization of a whole blood assay for quantifying myeloid-derived suppressor cells. *J Immunother Cancer* **7**, 230 (2019).
11. Shirasuna K., *et al.* Correlation analysis of the proportion of monocytic myeloid-derived suppressor cells in colorectal cancer patients. *PLoS One* **15**, e0243643 (2020).
12. Yang Z., Guo J., Weng L., Tang W., Jin S., Ma W. Myeloid-derived suppressor cells-new and exciting players in lung cancer. *J Hematol Oncol* **13**, 10 (2020).
13. Cossarizza A., *et al.* Guidelines for the use of flow cytometry and cell sorting in immunological studies (second edition). *Eur J Immunol* **49**, 1457-1973 (2019).
14. Wolf M., Rasche M., Eyrich M., Schmid R., Reinhardt D., Schlegel P. G. Spontaneous reversion of a lineage

switch following an initial blinatumomab-induced ALL-to-AML switch in MLL-rearranged infant ALL. *Blood Adv* **2**, 1382-1385 (2018).

15. Tirtakusuma R., *et al.* Epigenetic regulator genes direct lineage switching in MLL/AF4 leukemia. *Blood* **140**, 1875-1890 (2022).
16. Bataller A., *et al.* Myeloid lineage switch in KMT2A-rearranged acute lymphoblastic leukemia treated with lymphoid lineage directed therapies. *Haematologica* **109**, 293-297 (2024).

1 **Responses to the Reviewer's comments**

2 **Reviewer #1:**

3 I applaud to the authors for the efforts they spent addressing the concerns I raised in the initial review. They have
4 done a great job and answered most points. Here are a few additional suggestions:

5

6 **[Answer]**

7 We would like to express our sincere gratitude for your kind comments. Our responses to the additional suggestions
8 are as follows. Thank you again for your careful and constructive feedback.

9

10 1. p8 line 6-7: one of the potential explanations that LS AML is in cluster 2 is due to selection bias as they harbor
11 AFF1 fusion. The authors should at least mention it.

12

13 **[Answer]**

14 As you correctly point out, such possibility should be mentioned in the text. We added the explanation in the revised
15 manuscript (page 8, lines 9-10).

16

17 2. p8 line 21-23, related to the above concerns, the authors should compare LS AML with other clusters (not only LC
18 AML in cluster2) to see if similar conclusion can be drawn.

19

20 **[Answer]**

21 Thank you for the suggestion. As suggested, we have additionally compared LS AML samples with all LC AML
22 samples by GSEA (Supplementary Fig. 3a, Supplementary Data 3 and 4). The gene sets related to adaptive immunity
23 and antigen presentation were significantly downregulated in LS AML, similar to the previous comparison within
24 Cluster 2. Notably, gene sets related to antigen presentation were ranked even higher in this broader comparison.

25 We added the explanation in the revised manuscript (page 9, lines 3-7).

26

27 3. p10, line 9-14, regarding these variants identified by WES, I have major concerns. first, VAF of many variants are
28 very high including KRAS (KRAS or NRAS are typically subclonal with low VAF). second, TP53 variants seem
29 extremely common, which is contradictory to what we know in clinical patients that TP53 mutations are exceedingly
30 rare in MLL-r patients. Therefore, I highly suspect many of these variants are SNPs rather than pathogenic mutations.
31 The authors should re-examine the data and validate it seriously. Related to this, the whole discussion regarding TP53
32 needs to be removed.

33

34 **[Answer]**

35 We sincerely appreciate your insightful comments. In response to your concerns, we re-examined the data and
36 investigated copy number alterations in LS AML samples. We found that the high VAFs of TP53 in LS AML1 and
37 LS AML3, and CDKN2A in LS AML2, were attributable to copy number alterations. (Supplementary Fig. 3b).
38 However, no such alterations were observed for KRAS or other genes. We acknowledge that the frequency of TP53

1 mutations observed in our cohort is higher than generally expected in KMT2A-r patients. Nevertheless, we note that
2 in the recent Project EVOLVE study (Blood, 2025; <https://doi.org/10.1182/blood.2024026655>), TP53 mutations were
3 identified in 6 out of 45 patients (13.3%) with KMT2A-r LS AML, suggesting that TP53 mutations may be more
4 prevalent in this context than previously thought.

5 We appreciate the reviewer's important point regarding the interpretation of variants identified by WES. As many of
6 these variants may represent common SNPs rather than pathogenic mutations, and given the limitations of our current
7 dataset, we believe that it would be inappropriate to make any claims regarding their pathogenicity. In light of this,
8 we have removed all related interpretations from the text and now simply list the observed variants (page 10, lines
9 15–18). We have also deleted the corresponding discussion from the main text (page 19, line 12).

10
11 4. page 11, line 14-15, the statement of non-AML derived M-MDSCs is questionable. First, only one case showed a
12 few cells in monocytes; second, there is no genetic evidence showing these cells do not carry MLL fusion. So the
13 author should instead emphasize that M-MDSCs are largely leukemic cells. Relatedly, line 22-23 should be removed.

14
15 **[Answer]**

16 Thank you for raising important questions. In agreement with your comments, we have revised the sentence on
17 lines 14-15 (page 11, line 17) to emphasize that the M-MDSC-like cells are largely leukemic in origin. In addition,
18 we have removed the corresponding statement on lines 22-23 (page 12, line 1).

19
20 **Reviewer #2:**

21 All of my questions and concerns have been addressed. I have no further comments, and I believe the manuscript is
22 suitable for publication.

23
24 **[Answer]**

25 We were so encouraged by your comments during the revision. Thank you so much.

26 **Reviewer #3:**

27 The authors addressed successfully my comments.

28
29 **[Answer]**

30 We have learned a great deal from your insightful comments. Thank you again for your time and thoughtful
31 correspondence.

32
33 **Reviewer #4:**

34 The authors have sufficiently addressed my major concern of insufficient sample numbers by obtaining more
35 independent samples that strengthened their observations. I am comfortable with their conclusions now. Also, they
36 have sufficiently addressed my other concerns of MHC II expression, proper MDSC controls, attributing IFN γ
37 expression directly to T cells, and addressing caveats where appropriate.

38

1 **[Answer]**

2 I appreciate your kind instruction to strengthen our findings. We are sincerely grateful to you for your interest in our
3 research.

4

5

6

7

8

9

10

11

12

13

14

15

16

17

18

19

20

21

22

23

24

25

26

27

28

29

30

31

32

33

34

35

36

37

38